# Lead federated neuromorphic learning for wireless edge artificial intelligence

Helin Yang [1,2], Kwok-Yan Lam [2,3] ✉, Liang Xiao [1], Zehui Xiong [4], Hao Hu [5], Dusit Niyato [3] & H. Vincent Poor [6]

In order to realize the full potential of wireless edge artificial intelligence (AI), very large and diverse datasets will often be required for energy-demanding model training on resource-constrained edge devices. This paper proposes a lead federated neuromorphic learning (LFNL) technique, which is a decentralized energy-efficient brain-inspired computing method based on spiking neural networks. The proposed technique will enable edge devices to exploit brain-like biophysiological structure to collaboratively train a global model while helping preserve privacy. Experimental results show that, under the situation of uneven dataset distribution among edge devices, LFNL achieves a comparable recognition accuracy to existing edge AI techniques, while substantially reducing data traffic by >3.5× and computational latency by >2.0×. Furthermore, LFNL significantly reduces energy consumption by >4.5× compared to standard federated learning with a slight accuracy loss up to 1.5%. Therefore, the proposed LFNL can facilitate the development of brain-inspired computing and edge AI.

In recent years, with the rapid development of mobile computing and Internet of Things (IoT), billions of devices such as sensors, actuators, robots, and autonomous vehicles are connected, generating massive amounts of data[1]. Driven by this trend, a powerful technique termed edge artificial intelligence (AI), amalgamating edge computing and AI[2–8], has been proposed to enable devices on the edge of a network to locally analyze and process data without transferring collected data to a centralized server. Such capability not only facilitates data privacy preservation but also reduces data traffic and network latency. Moreover, unprecedented accuracies have been achieved by deep learning of neural networks trained for speech recognition, image and video classification, and object detection in edge AI[3–7]. Despite these benefits, edge AI still faces the following two fundamental challenges. Firstly, modern AI-based algorithms depend intrinsically on sophisticated learning methods[7], and more importantly on sufficiently rich training datasets[9,10]. Thus, the limited sizes of local datasets available to edge devices inevitably make the task of training usable AI models almost impossible[9,10]. Secondly, machine learning algorithms are generally computing intensive and energy-demanding, which hampers energy-constrained edge devices from training/analyzing data locally[2,3,11,12].

One potential technique to address the first challenge is federated learning (FL)[13,14]. In FL, as reported[2,4,15–17], multiple collaborative devices locally train a machine learning model (i.e., each with its own data, in parallel) without uploading raw data to a server. In this context, the devices only upload parameters (or gradients) to a central server for global model aggregation. Then, the updated model parameters are sent back to devices for the next training epoch, and the process is repeated until convergence. FL not only enables edge AI to achieve a comparable model quality to centralized learning, but also reduces data traffic and helps preserve data privacy. For these

[1]Department of Information and Communication Engineering, School of Informatics, Xiamen University, Xiamen, China. [2]Strategic Centre for Research in Privacy-Preserving Technologies and Systems, Nanyang Technological University, Singapore, Singapore. [3]School of Computer Science and Engineering, Nanyang Technological University, Singapore, Singapore. [4]Pillar of Information Systems Technology and Design, Singapore University of Technology and Design, Singapore, Singapore. [5]Department of Electrical and Electronic Engineering, Nanyang Technological University, Singapore, Singapore. [6]Department of Electrical and Computer Engineering, Princeton University, Princeton, NJ, USA. ✉e-mail: kwokyan.lam@ntu.edu.sg

reasons, FL has recently been applied in privacy-sensitive medical applications[10,18–20], e.g., medical image classification[18]. Considering the central coordinator in FL, all clients/devices are required to trust the central server and the training speed is limited by the heterogeneity of edge devices[21]. To address this issue, decentralized FL has been proposed[20–25], where model parameters are exchanged only among interconnected devices without using a central server. Even so, repeatedly cycling model aggregation among devices results in increased training latency[26–28]. Furthermore, even if centralized or decentralized FL provides a solution for privacy-enhancing and reliable model training under insufficient datasets at edge devices, model training based on deep learning can consume a significant amount of

energy, further hindering application of decentralized FL in energy-constrained edge devices.

As noted above, standard deep learning algorithms, e.g., multilayer artificial neural networks (ANNs) and convolutional neural networks (CNNs), are generally power-hungry[29–32]. To address this challenge, inspired by biological neurons, spiking neural networks (SNNs)[33,34] have been proposed and explored as a promising neuromorphic computing solution for the implementation of AI algorithms in edge devices due to their low energy consumption. SNNs simulate the electrical activity of human-brain systems and operate with continuous spatio-temporal dynamics and discrete spike events using Integrate-and-Fire (IF) or Leaky IF (LIF) neuron units[35]. Owing to the

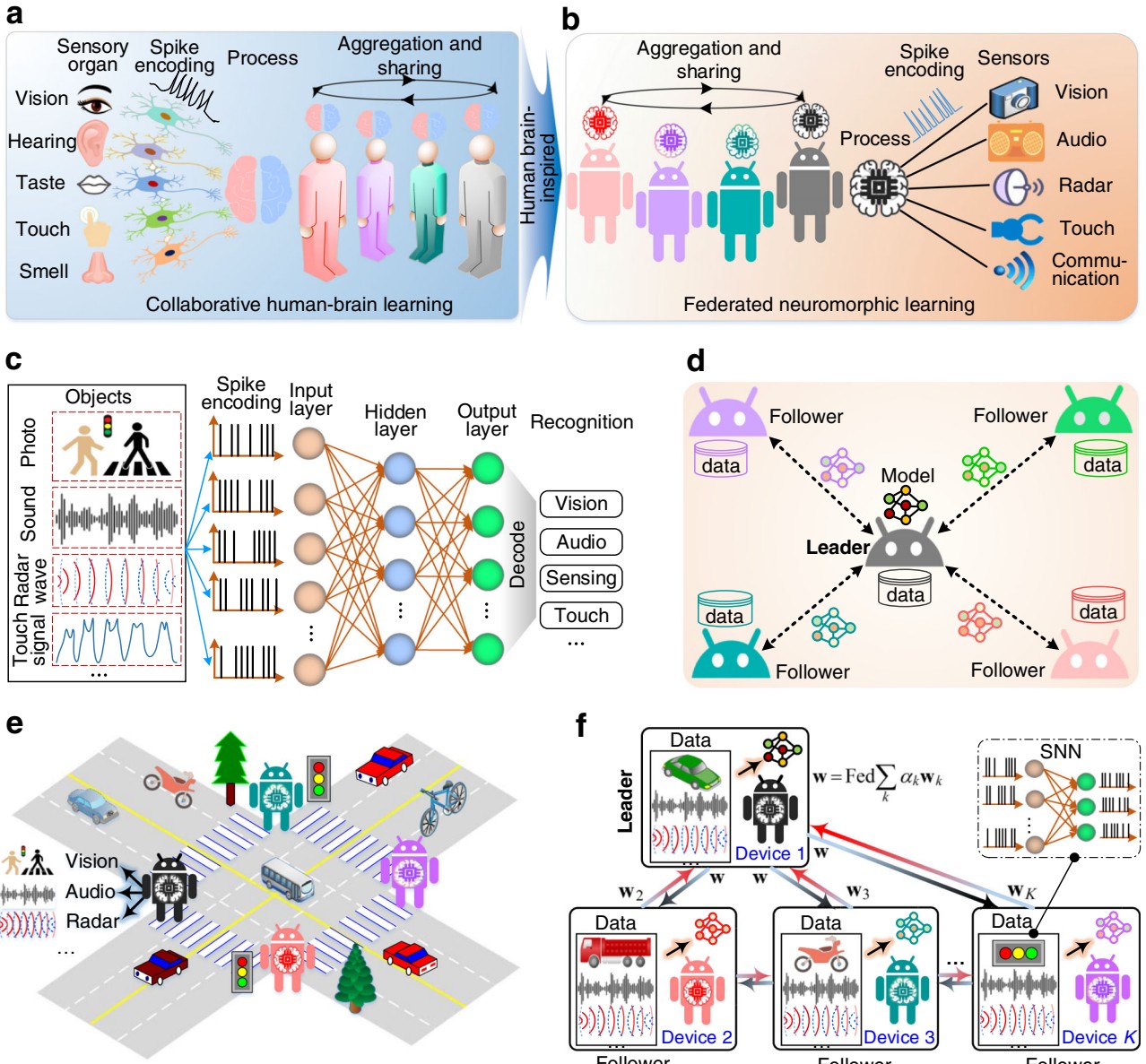

**Fig. 1 | Schematic diagram of the lead federated neuromorphic learning system. a** Schematic of a social learning network, where each human uses five sensory organs to interact with the outside environment via neural networks. The humans in a group exchange learned knowledge with each other for better recognition. **b** Inspired by the collaborative learning system in (**a**) a federated neuromorphic learning system is introduced to perform model aggregation from edge devices in a group. The integration between devices and the external environment is handled by using sensors (e.g., cameras, microphones, radars, and touch sensors). **c** The structure of an SNN which is adopted to perform

neuromorphic computing for edge devices. **d** The principle of LFNL without a central server, where one device is selected as a leader to manage model aggregation in a group. **e** An example situation of multiple humans crossing a vehicular road, where multiple edge devices can observe, hear, and sense traffic objects. **f** Illustration of LFNL-based traffic recognition. The leader (device) leads other followers (devices) to train their own local neuromorphic models independently, and it collects local model parameters ($w_2$, $w_3$, ..., $w_K$) to perform model aggregation before broadcasting **w** to the followers for the next local training. The exchange of local and global parameters repeats until convergence.

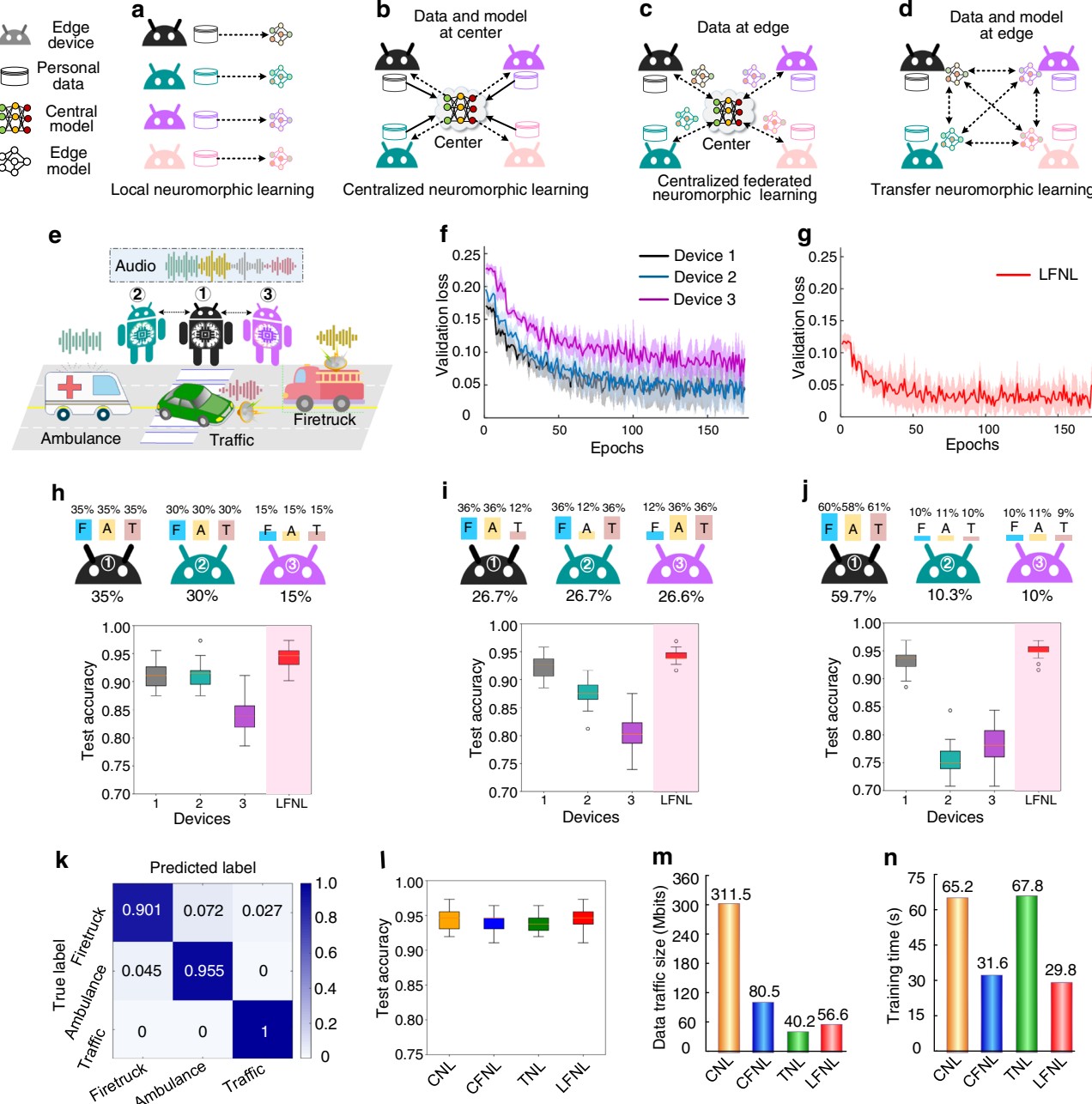

**Fig. 2 | LFNL for audio recognition on a road. a** Principle of local neuromorphic learning (LNL) with data and model on the devices. **b** Principle of centralized neuromorphic learning (CNL) with data and model being stored at the central server. **c** Principle of centralized federated neuromorphic learning (CFNL) with data being kept on the devices, and local model parameters being uploaded to the central server for model aggregation. **d** Principle of transfer neuromorphic learning (TNL) with data being kept on the device side, and each device trains its model and then passes it to the next device for training, cyclically repeating the process. **e** An example of audio recognition on a road, including firetruck, ambulance and general traffic sounds. **f, g** Validation loss curves for three locally training devices and LFNL. **h–j** Box plots show test accuracy performed for three locally training devices and LFNL with uneven distributions of training dataset (F: firetruck sound class, A: ambulance sound class, T: general traffic sound class). The training dataset distributions of three sound classes for three devices are shown at the top of these figures. **k** Confusion matrix for the test set in LFNL after training. **l–n** Test accuracy, data traffic, and training latency comparisons for different learning methods, respectively.

inherent parallelism of binary spike-based sparse computing over time steps, SNNs promise fast, sparse, and energy-efficient information processing[36–41]. Furthermore, several attempts have been made to combine SNNs with FL[42,43] to improve learning capability, energy efficiency, and privacy preservation, but model parameters are still aggregated by a central server.

In this article, we propose lead federated neuromorphic learning (LFNL), a decentralized brain-inspired computing method based on

SNNs, enabling multiple edge devices to collaboratively train a global neuromorphic model without a fixed central coordinator. In particularly, we present a leader election scheme to elect one device with high capability (e.g., computation and communication capabilities) as a leader to manage model aggregation. This approach can effectively accelerate the convergence of federated learning and defend against model poisoning attacks. Experimental results demonstrate that LFNL achieves high classification accuracies of 94.3%, 95.6% and 94.7% on

audio, visual and radar signal recognition tasks with uneven dataset distribution among devices. Such high accuracies are approximately equivalent to those of centralized learning and significantly outperform local learning. LFNL also substantially reduces data traffic and computational latency compared to centralizing learning. The results further verify that LFNL yields approximately the state-of-the-art accuracy (up to 1.5% loss) with significant energy consumption reduction (~4.5×) compared to standard federated learning methods. LFNL promises several important benefits for edge AI compared to existing computing paradigms, including privacy enhancement, low computational latency, data traffic reduction, energy efficiency, and robustness. As such, LFNL is envisioned to significantly boost the development of brain-inspired computing and edge AI.

## Results

### Construction of lead federated neuromorphic learning

In order to enable edge devices to perform computing with low energy consumption, low latency, and high-accuracy recognition with privacy-enhancement, we developed an LFNL system, as shown in Fig. 1. Figure 1a shows a schematic diagram of a collaborative human social system. Each human uses five general sensory organs to observe analog stimulus from an outside environment, and then the stimulus is transformed into a spike signal using specialized neurons which are then processed by the human's brain. Each human builds a corresponding knowledge model, and then shares the model with others to create an optimized knowledge model for better recognition. Inspired by this, a federated neuromorphic learning system is introduced for edge AI (Fig. 1b), where the edge devices are equipped with cameras (vision), microphones (hearing), radars (object sensing), pressure sensors (touch), and radio-frequency signal detectors (wireless communication). These sensors adopt SNNs as a neuromorphic processor to convert detected information into spike signals. The structure of SNNs with Meta-Dynamic Neurons (MDNs)[32] is illustrated in Fig. 1c, and the inputs of SNNs are discrete spikes which are encoded from object analog signals (vision, audio, radar, etc.)[32]. The signals in input, hidden and output layers of SNNs are all spike trains (see Methods).

LFNL is implemented with a leader and a number of followers in a group (Fig. 1d), and the learning model parameters are shared and exchanged via distributed networks with each device training its model independently on local data. Note that a device with high computation, communication, and energy supply capabilities is elected as the leader to effectively manage model aggregation and accelerate the federated learning process (the leader election protocol and performance evaluation are discussed in Supplementary Figs. 1, 2). To better illustrate the concept of LFNL for edge AI, we consider a scenario in which several edge devices cross a road as an example (Fig. 1e). In a social network, humans share their learning knowledge with each other to provide better object recognition, and one of them acts as a leader to lead the group members to better explore, learn, and adapt to the physical world. Inspired by the human-like learning functionalities, LFNL realizes object recognition by training or evaluating spike signals from auditory, vision, and radar systems using neuromorphic learning (Fig. 1e). The objective of the leader is to aggregate the uploaded local neuromorphic model parameters ($w_2$, $w_3$, ..., $w_K$) from the followers. All followers only need to send their local model parameters to the leader instead of uploading their raw data. After aggregating the model parameters at each global epoch, the updated global parameter $w$ will be sent to followers for the next training epoch. Further details about the LFNL can be found in Methods.

### Application to audio recognition

We first tested the audio recognition capability of LFNL, and selected a traffic sound dataset (https://www.kaggle.com/vishnu0399/emergency-vehicle-siren-sounds) for performance evaluation. To compare LFNL with other techniques, we introduce four benchmark methods based on SNNs, each of which incorporates some, but not all, of the benefits of LFNL. Briefly, these are as follows. Local neuromorphic learning (LNL) enables each device to locally train its model without sharing raw data with other devices (Fig. 2a). Centralized neuromorphic learning (CNL) uses a central server to collect datasets of all devices for global model training (Fig. 2b). Venkatesha et al.[43] designed a centralized federated neuromorphic learning (CFNL) method for training decentralized and privacy promoting SNNs, where a central server is used to perform global SNN model aggregation. CFNL can keep the raw data on the devices (Fig. 2c), and only the local model parameters need to be uploaded to the central server for model aggregation which enhances the global accuracy. However, CFNL still relies on a centralized structure. Transfer neuromorphic learning (TNL) keeps data on the devices (Fig. 2d), and each device trains its model and then passes it to the next device for training, repeating the process cyclically. However, the devices train in sequence rather than in parallel, leading to a longer training latency[27,28].

We implemented the experiments on several Raspberry PI 4Bs, a Raspberry PI 3B+ and one laptop (see Methods). In the benchmark LFNL, the SNN has 128-2000-3 neurons corresponding respectively to the input-hidden-label layers. For the traffic sound dataset, total of 600 sound samples were used with three classes, including firetruck, ambulance, and general traffic sound samples with each having 200 samples. 80% of the sound samples were used for training, and the remaining 20% of their samples were used for validation and testing.

As the validation loss is widely used to measure the quality of training capability, we considered this metric versus training epochs for three locally training devices and LFNL. Since device 3 has insufficient training samples (Fig. 2f), it has a higher validation loss than those of the other two devices having more training samples, leading to a lower test accuracy of 84.2% (Fig. 2h). Device 1 and device 2 achieve test accuracies of 91.3% and 91.4% (Fig. 2h), respectively. However, by applying LFNL, the system achieves a faster training convergence speed (Fig. 2g) and a higher test accuracy of 94.3% (Fig. 2h) than those of the three locally training devices. In general, AI performs well when the training data is sufficient, for example, the training and test accuracy of device 1 and device 2 are quite good. We further examined the stability and robustness of LFNL with an uneven dataset distribution of the three classes on the three devices. As depicted in Fig. 2i, as each device has significantly insufficient training samples on one class dataset, the overall test accuracy of the three locally training devices decreases substantially compared to Fig. 2h. Similarly, this phenomenon also happens with the uneven distribution of datasets on device 2 and device 3 (Fig. 2j). However, LFNL still maintains a high test accuracy of 95% and significantly outperforms locally training devices. Moreover, the LFNL results do not deteriorate when we divided the training samples into six smaller parts for six devices (Supplementary Fig. 3). In addition, we find that LFNL can effectively defend against model poisoning attacks (Supplementary Fig. 4).

We further compared the performance of LFNL with other learning methods (Fig. 2l–n). As depicted in Fig. 2l, LFNL not only achieves a similar test accuracy to CNL, but also significantly reduces both data traffic size by >3.5× (Fig. 2m) and training latency by ~2.0× (Fig. 2n). Both CFNL and LFNL train the learning model in parallel, and have approximately similar test accuracies (Fig. 2l) and training latencies (Fig. 2n), but the former needs a central server for model aggregation and it also has higher data traffic (Fig. 2m). Unlike LFNL, TNL trains the model sequentially rather than in parallel and has lower data traffic than that of LFNL (Fig. 2m), but it requires much longer computational latency (Fig. 2n). The results shown in Fig. 2l–n

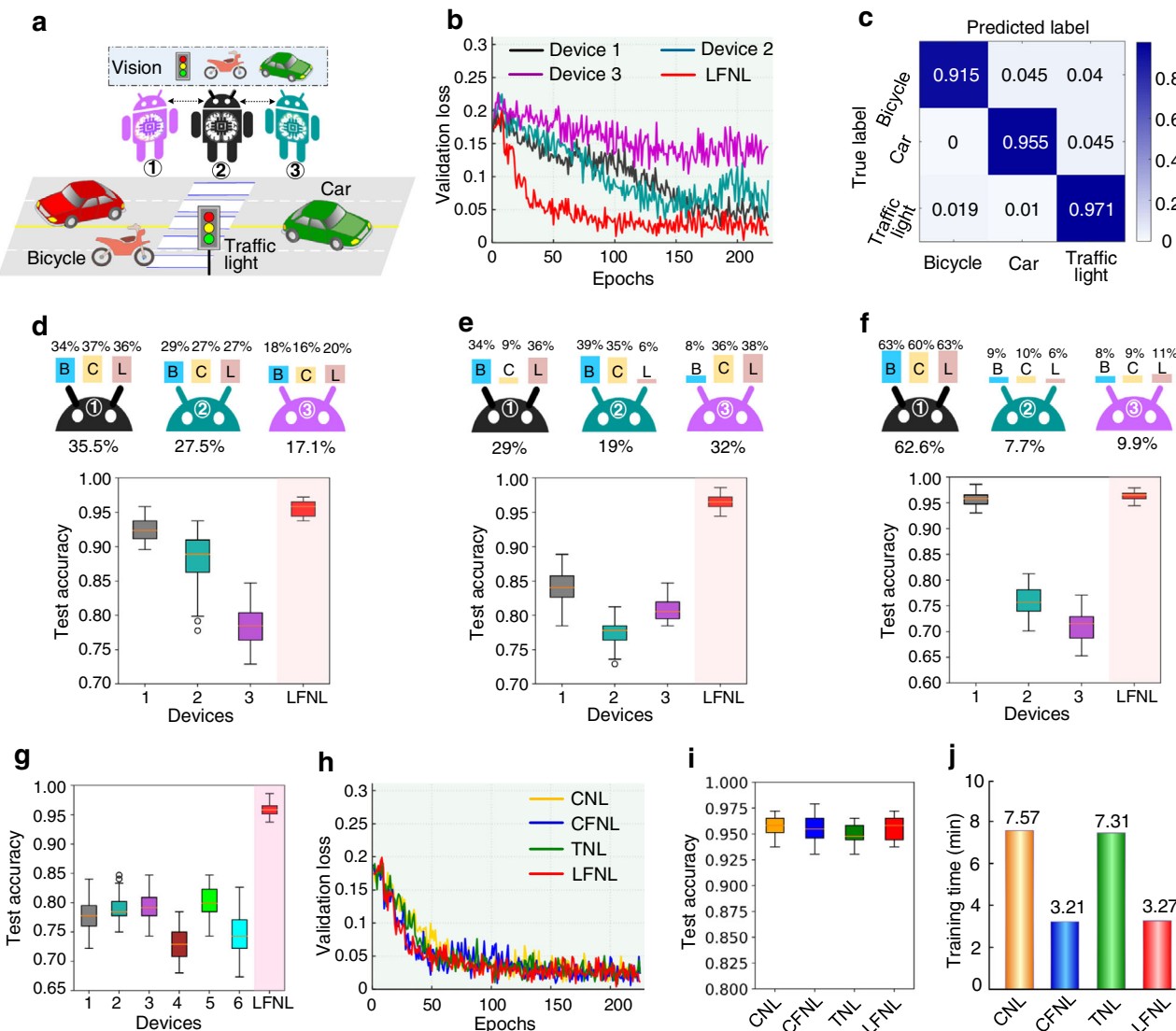

**Fig. 3 | LFNL for visual recognition on a road. a** An example of visual recognition (image classification) on a road, including bicycle, car, and traffic light image classes. **b** Validation loss curves for three locally training devices and LFNL. **c** Confusion matrix for the test set in LFNL after training. **d–f** Box plots show test accuracy performed for three locally training devices and LFNL with uneven distributions of training dataset (B: bicycle image class, C: car image class, L: traffic light image class). **g** Illustration of box plots showing test accuracy performed for six locally training devices and LFNL. **h–j** Validation loss curves, test accuracy and training latency performance comparisons for different learning methods, respectively.

strongly support the conclusion that LFNL is more suitable for edge AI in terms of recognition accuracy, data traffic, latency and privacy-enhancement factors.

## Application to visual recognition

Next, we applied LFNL to implement visual recognition, as illustrated in Fig. 3a. In the benchmark LFNL, the SNN has 1728-2500-3 neurons corresponding respectively to the input-hidden-label layers. For the traffic image dataset (https://www.kaggle.com/vishnu0399/emergency-vehicle-siren-sounds, https://www.kaggle.com/hj23hw/pedestrian-augmented-traffic-light-dataset), in total, 872 images are used with three classes, including 160 bicycle images, 205 car images, and 507 traffic light images. 80% of the images were used for training, and the remaining 20% of the images were used for validation and testing.

Figure 3b illustrates the validation loss curves of three locally training devices and LFNL. The training dataset distributions of three traffic types (i.e., bicycle (B), car (C) and traffic light (L)) for three devices are shown at the top of Fig. 3d. Due to the insufficient training images, both device 2 and device 3 overfit quickly and result in unstable training (Fig. 3b), achieving low test accuracies of 88.0% and 78.5% (Fig. 3d), respectively. Compared with the three locally training devices, LFNL overcomes the local overfitting, exhibits smoother and significantly obtains a smoother and faster convergence (Fig. 3b), and achieves a higher test accuracy of 95.6% (Fig. 3d). We further examined the stability and robustness of LFNL with a significantly uneven and insufficient dataset distribution. From Fig. 3e–g, we find that the overall test accuracy of locally training devices with uneven and insufficient datasets declines significantly, whereas the LFNL results do not deteriorate. Furthermore, LFNL still robustly achieves a high classification accuracy under random image rotation angles (Supplementary Fig. 5).

We also considered the performance comparisons for different learning methods (Fig. 3h–j). As shown in Fig. 3h, LFNL achieves similar training convergence and validation loss values to those of CNL, CFNL and TNL. In addition, it also obtains a comparable test accuracy to other learning methods (Fig. 3i). However, by using our LFNL method, the training latency is significantly reduced in

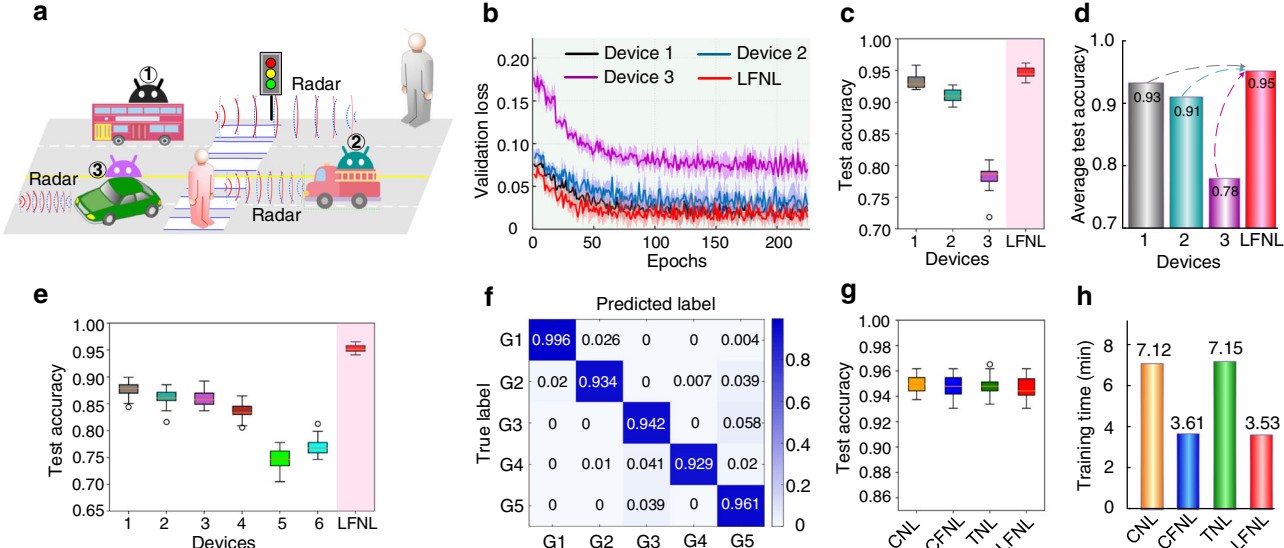

**Fig. 4 | LFNL for radar signal recognition on a road. a** An example of radar signal recognition on a road, which has five radar gesture classes. **b** Validation loss curves for three locally training devices and LFNL. **c**, **d** Box plots show test accuracy and the histogram shows the average test accuracy performed for three locally training devices and LFNL after training. **e** Illustration of box plots showing test accuracy comparison to CNL and TNL (Fig. 3i). We note that both CNL and performed for six locally training devices and LFNL, where the training dataset is divided into six smaller parts for them. **f** Confusion matrix for the test set in LFNL after training. **g**, **h** Test accuracy and training latency performance comparisons for different methods, respectively.

comparison to CNL and TNL (Fig. 3i). We note that both CNL and CFNL rely on a central structure which increases the data traffic for edge AI. In these results (Fig. 3b–j), LFNL significantly outperforms individual devices regardless of how uneven the data distributions are, and its recognition capability is close to that of the centralized learning method.

### Application to radar signal recognition

This section tests the radar signal recognition capability of LFNL, where we simulate a situation (Fig. 4a) in which devices (e.g., vehicles) use radar systems to recognize human gestures on a road, for applications such as recognizing when a person crosses a road, hails a taxi, or signals for a bus to stop. In total, 1695 five-class radar gesture samples[31] were adopted for classification evaluation, where 80% and 20% of the samples were used for training and testing, respectively. In the benchmark LFNL, the SNN has 4800-1000-5 neurons corresponding respectively to input-hidden-label layers. Figure 4b presents the validation loss curves for three locally training devices and LFNL, where the dataset distribution for the three devices are set as 39.5%, 27.8% and 12.7%, respectively. Similarly to previous results, due to insufficient training samples, device 3 achieves higher validation loss values than those of the other two devices, leading to a lower test accuracy of 78.3% (Fig. 4c). Fortunately, this problem can be mitigated using LFNL, as illustrated in Fig. 4c, d, in which it is seen that the accuracies of the three locally training devices are obviously improved from 93.2%, 91.1% and 78.3%, respectively, to 94.7%. Further, we divided the training samples into six parts for six devices with each having a smaller training dataset, in particular, the dataset sizes at the six devices were set as 17.7%, 17.7%, 14.8%, 11.8%, 8.9%, and 9.1%, respectively. As shown in Fig. 4e, the overall test accuracy of the six locally training devices significantly decreases, especially the performance of device 5 and device 6, because they have very small training samples, whereas the LFNL results do not deteriorate. The confusion matrix of five-label recognition is provided in Fig. 4f. For each gesture class, only very few samples are misclassified into other classes. Similar to the corresponding results in Figs. 2 and 3, the test accuracy of LFNL is

still equivalent to other learning methods (e.g., CNL and TNL), as shown in Fig. 4g. However, LFNL significantly reduces the training latency compared with CNL and TNL (Fig. 4h), and it does not need a central server compared with CNL and CFNL.

### Analysis of recognition accuracy and energy consumption

This section compares the accuracy and energy consumption (Fig. 5) of LFNL-SNN with those of standard lead federated learning based ANN (LFL-ANN), in order to illustrate the energy efficiency of LFNL. Note that both SNNs and ANNs have the same learning structure for fair comparison. The details of energy consumption analysis can be found in Methods.

In Fig. 5a, d, and g, we find that the accuracy of LFNL-SNN suffers a slight loss (up to 1.5%) compared to LFL-ANN. In ANN, the neurons use high-precision activation and continuous values, and propagate signals only in the spatial domain, whereas the signals in SNNs are spike trains coded in binary events instead of continuous activation and each spiking neuron exhibits rich dynamic behavior. Thus, LFNL-SNN generally has more temporal versatility but slightly lower recognition accuracy (Fig. 5a, d, g) compared to LFL-ANN[36–43]. For example, as illustrated in Fig. 5e, the visual recognition accuracy of LFNL-SNN is 94.3%, only slightly lower than that of LFL-ANN with the accuracy of 95.8% in this scenario.

Figure 5c, f, h show the estimated energy consumption of ANN and SNN models trained on the audio, vision and radar datasets for three devices, respectively. From these figures, compared with LFL-ANN, the significantly larger energy saving gains achieved can be attributed to the sparsity obtained with event-driven spike trains in SNNs. For example, for visual recognition illustrated in Fig. 5f, the energy consumption of LFL-ANN is 13.85 μJ, whereas that of LFNL-based SNNs is 2.92 μJ which is 4.75× in reduction. Thus, we see that, in using LFNL-SNN for edge AI, the energy needed for computation is significantly reduced while he accuracy is largely preserved compared with its ANN-based counterpart.

### Application in complex and high-dimensional datasets

In addition to evaluating performance on audio, visual, and radar signal recognition tasks, LFNL is expected to be effective and robust

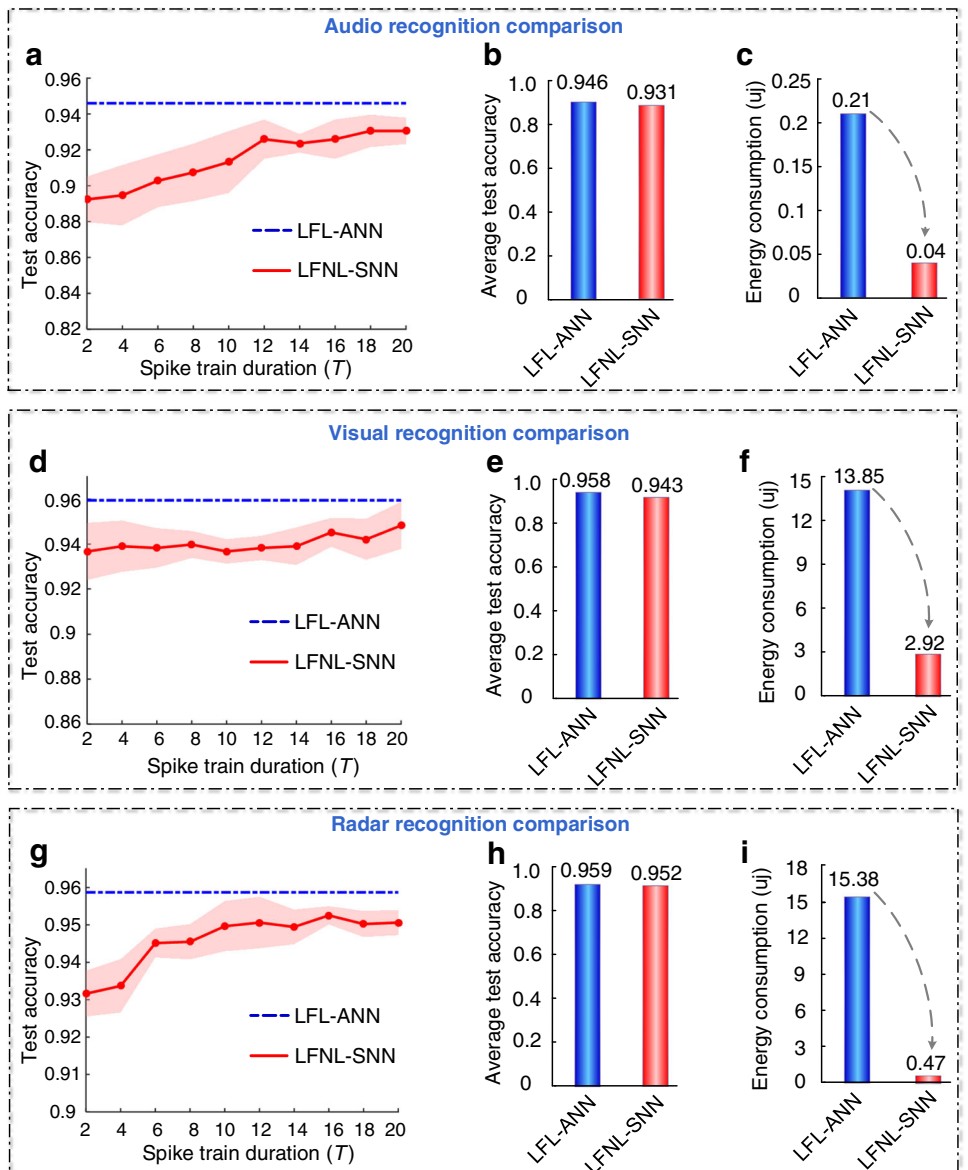

**Fig. 5 | Recognition accuracy and computation energy comparisons between ANNs and SNNs trained on the audio, vision and radar datasets. a** Audio classification accuracy comparison between LFNL-SNN and LFL-ANN versus the spike train duration $T$ of the operation of SNNs on the traffic sound dataset. Both SNNs and ANNs have the same learning structure with 128-500-3 neurons (for input-hidden-label layers). **b, c** Average test accuracy and energy consumption comparisons when the spike train duration $T$ is 15. **d** Visual classification accuracy comparison between LFNL-SNN and LFL-ANN versus $T$ of SNNs on traffic image dataset. Both SNNs and ANNs have the same learning structure with 1728-2500-3 neurons. **e, f** Average test accuracy and energy consumption comparisons when the spike train duration $T$ is 15. **g** Radar gesture classification accuracy comparison between LFNL-SNN and LFL-ANN versus $T$ of SNNs on radar gesture dataset. Both SNNs and ANNs have the same structure with 4800-1000-5 neurons. **h, i** Average test accuracy and energy consumption comparisons when the spike train duration $T$ is 15.

on complex and high-dimensional datasets. Therefore, we further applied LFNL to evaluate the classification accuracy on CIFAR10 and CIFAR100 datasets[44]. Note that we used the github repository[43] provided by Venkatesha et al.[43] to run the experiments and obtain the CFNL results described below.

Figure 6 shows the experimental results for LFNL and CFNL methods on the CIFAR10 and CIFAR100 datasets using the VGG9 training model[44], where different parameters (i.e, participating device configurations, non-IID distribution and gradient noise) are considered in the performance evaluation. Here, we use $P/N$ to show the device split, where $N$ means that the dataset is divided into $N$ parts for $N$ devices, and $P$ represents the number of participating devices selected for model aggregation in each global round[43]. From Fig. 6a, b, we observe that the classification

accuracy of the two methods gradually decreases as the number of devices increases. The reason for this is that the training dataset is divided among more devices which degrades the learning capacity with insufficient local training data. We further evaluated the classification accuracy under different levels non-IID-ness of the data, where the Dirichlet distribution with concentration parameter $\alpha$ is used to obtain non-identical datasets[45,46]. Note that as the value of $\alpha$ decreases, the class composition becomes more skewed and the degree of the the non-identicality of the data becomes more pronounced. Figure 6c, d show that even though the data becomes more non-IID as the parameter $\alpha$ decreases, the classification accuracies of both LFNL and CFNL do not decline significantly when $\alpha \geq 1$. However, when $\alpha < 1$, the non-IID-ness of the data tends to be more skewed as $\alpha < 1$ decreases, and the

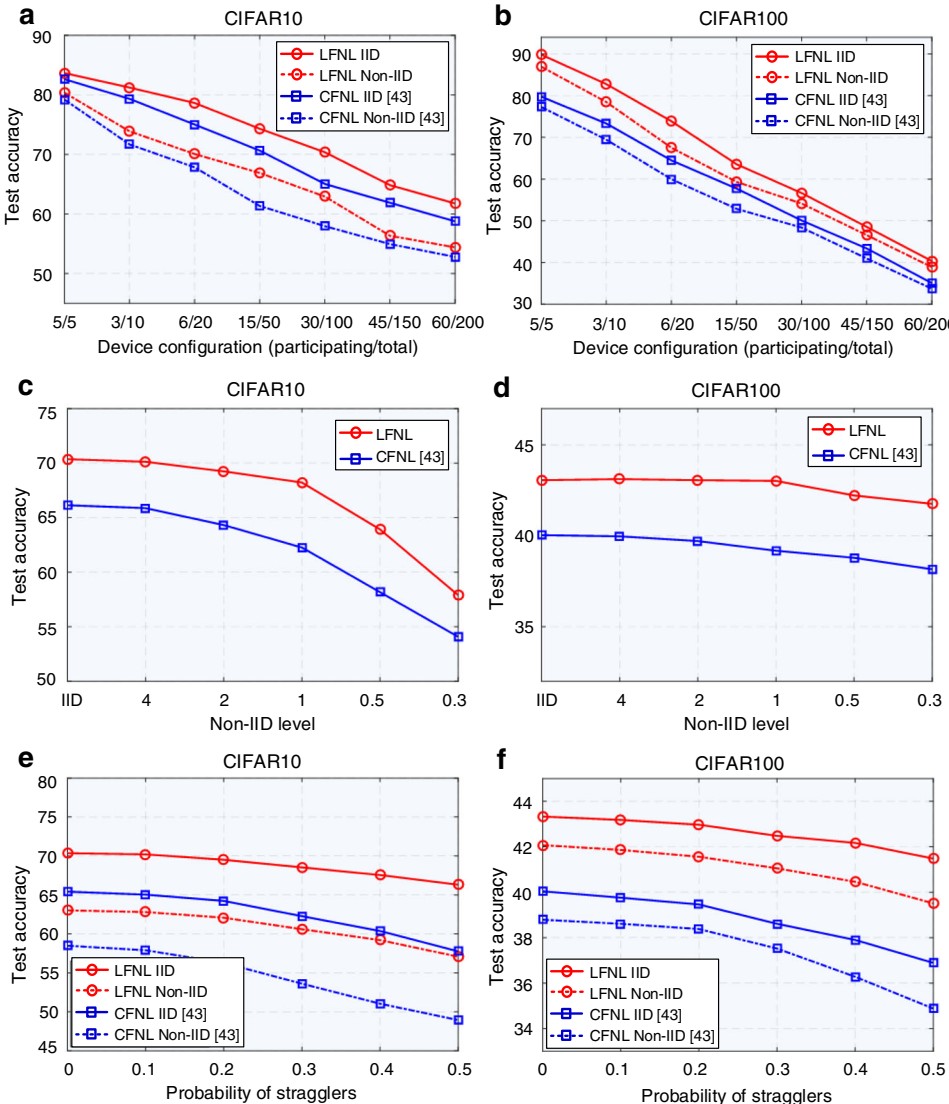

**Fig. 6 | Classification accuracy evaluation on CIFAR10 and CIFAR100 datasets.** **a**, **b** Performance evaluation of LFNL vs CFNL under different device configurations. **c**, **d** Impact of the non-IID data factor on the classification accuracy of LFNL vs CFNL when the training dataset is divided among 100 devices and 30 devices participating in each global round. **e**, **f** Effects of the probability of stragglers on the performance of LFNL vs CFNL when the training dataset is divided among 100 devices and 30 devices participating in each global round.

training model diverges significantly. Finally, there is a steady decrease in the classification accuracy of LFNL and CFNL. Due to the the unreliability of wireless environments, some devices will fail to communicate their model parameters to the central server or leader, and we capture this by considering the probability of such stragglers. As shown in Fig. 6e, f, for both the CNFL and LFNL methods, the negative effect of the probability of stragglers on the test accuracy is not significant in the case of IID training data. However, the accuracy decreases more significantly in the non-IID case as the probability of stragglers increases. From Fig. 6a–f, we can see that LFNL still achieves a higher classification accuracy than that of CNFL under different parameters on both CIFAR10 and CIFAR100 datasets, as the LFNL elects the leader to be the device with higher overall channel quality which reduces the probability of stragglers.

## Discussion

In this paper, we have proposed a lead federated neuromorphic learning method for edge AI, namely LFNL, integrating brain-inspired neuromorphic computing and federated learning in the domain of human-like machine intelligence. LFNL enables edge devices to collaboratively train a global reliable model while enhancing privacy without a central server, in the presence of uneven and insufficient training data on edge devices. Owing to the decentralized federated learning and parallel training structures, LFNL becomes an effective alternative to the centralized data sharing paradigm across edge devices without relying on any central server, and thus significantly reduces the heavy data traffic, enhance data privacy and decreases training latency compared to existing centralized learning methods. Moreover, with the implementation of spike-based processing features, our proposed LFNL can substantially reduce energy consumption, which makes LFNL particularly suitable for energy-constrained edge devices.

The advantages of LFNL have been demonstrated experimentally in a series of benchmark comparisons on audio, visual and radar signal recognition tasks under uneven dataset distributions. We have seen that LFNL achieves an inference accuracy of more than 94% for

each task, and it significantly outperforms the locally training method and obtains a comparable recognition accuracy to centralized learning without causing heavy data traffic, as shown in Figs. 2–4. Due to the spike activation driven nature of LFNL, the method requires a finite number of training time steps $T$ to optimize LFNL-SNN and obtains slight lower classification accuracy than that of the standard federated learning-based ANNs. However, it can significantly reduce the energy consumption for energy-constrained devices (Fig. 5). Due the scalability of LFNL, such a higher classification accuracy is still achieved compared with the existing federated learning framework on larger and higher-dimensional datasets (CIFAR10 and CIFAR100 datasets) (Fig. 6).

In brief, LFNL offers a powerful mechanism for democratizing the use of neuromorphic learning in the domain of human-like machine intelligence. Owing to the aforementioned benefits and advantages, LFNL can effectively deploy deep learning of neural networks for resource-constrained edge devices with various practical applications, such as speech recognition, image and video classification, smart sensing, health monitoring and multi-object detection in edge AI. It also has the potential to implement deep learning on large-scale scientific/industrial systems, for example, autonomous instruments, autonomous vehicles and mission critical diagnostics. This gives us confidence that LFNL can contribute significantly to the development of brain-inspired computing and edge AI.

## Methods
### SNN model
In order to enable SNNs to achieve better learning efficiency and generalization for object recognition/classification in LFNL, SNNs combined with an MDN architecture[32] is used in our work. The MDN is designed with meta neurons including the first-order and second-order dynamics of membrane potentials, as well as the spatial and temporal meta types supported by hyper-parameters[32].

Figure 1c shows a typical SNN architecture with LIF neurons. The spiking neurons in SNNs communicate with each other using spike trains coded in binary events (1: spike, 0: no spike) in a temporal domain over a given number of time steps $T$, referred to as spike train duration. In this work[35], LIF[35] is applied to perform standard first-order dynamic spike neurons. The dynamic includes only up to an attractor. The dynamic behavior of the $i$-th spike neuron using LIF is characterized by

$$\tau \frac{dU_i(t)}{dt} = \beta U_i(t) + C(t), \tag{1}$$

where $U_i(t)$ denotes the the membrane potential and $\tau$ is the time constant for $U_i(t)$. $C(t)$ is the input synaptic current (the weighted summation of pre-spike events) at time $t$ and can be written as

$$C(t) = \sum_{i=1}^{M} w_{i,j} \sum_{n=1}^{N} V_i(t - t_n), \tag{2}$$

where $V_i(t - t_n)$ is the spike event from the current neuron $i$ to its pre-neuron $j$, $w_{i,j}$ is the synaptic weight between $V_i(t - t_n)$ and $N$ denotes the number of neurons. In this context, the dynamic behavior of spike neurons in the first-order is given by &

$$\begin{cases} \tau \frac{dU_i(t)}{dt} = \beta U_i(t) + C(t), \\ S_i(t) = 1 \qquad\qquad \text{and } U_i(t) + U_{re}, \text{ if } U_i(t) \geq U_{th}, \\ S_i(t) = 0, \text{ if } U_i(t) < U_{th}, \end{cases} \tag{3}$$

where $S_i(t)$ is the output of the $i$-th neuron at time $t$, and $U_{re}$ and $U_{th}$ are the reset potential and firing threshold (resting potential), respectively. In Eq. (3), the spike event $S_i(t)$ can be obtained from the

value of the membrane potential $U_i(t)$, e.g., $S_i(t) = 1$ when $U_i(t) \geq U_{th}$, and otherwise, $S_i(t) = 0$.

The dynamic behavior of the $i$-th second-order neuron with MDNs can be written as[32]

$$\begin{cases} \tau \frac{dU_i(t)}{dt} = U_i^2(t) - U_i(t) - H_i(t) + C(t), \\ \frac{dO_i(t)}{dt} = \eta_a(\eta_b U_i(t) - H_i(t)), \\ S_i(t) = 1, \qquad\qquad U_i(t) = \eta_c \text{ and } H_i(t) = H_i(t) + \eta_d, \text{ if } U_i(t) \geq U_{th}, \\ S_i(t) = 0, \text{ if } U_i(t) < U_{th}, \end{cases}$$
$$\tag{4}$$

where $H_i(t)$ denotes a resistance value simulating hyperpolarization which is tapped to charge the activation and inactivation of currents, and $\eta_a$, $\eta_b$, $\eta_c$, and $\eta_d$ are dynamic parameters which are used to distinguish the different second-order dynamics of the membrane potential in the SNN. From Eq. (4), we can observe that the attractor of $U_i(t)$ is determined by $H_i(t)$ and $C(t)$[32].

In an SNN, the loss function is used to evaluate the mean square error between output firerates $S_i(t)$ and classification labels $Y_i$, which is given by

$$Loss = \sum_{i=1}^{M} \left( \frac{1}{T} \sum_{t=1}^{T} S_i(t) - Y_i \right)^2. \tag{5}$$

Since SNNs involve non-differentiable functions, the widely-used gradient-based back propagation (BP) cannot be used directly to train them. Rather, an approximate BP technique to train the SNNs model in LFNL. Thus, an approximate BP trick technique[32,36] is used to train SNNs using a pseudo differential gradient give by

$$G = \begin{cases} 1, \text{ if } |U_i(t) - U_{th}| < U_{tar}, \\ \qquad 0, \text{ otherwise}, \end{cases} \tag{6}$$

which is used to dynamically update the synaptic values in SNNs, and where $U_{tar}$ is the range of membrane potential between input. The learning structure is trained on a sequence of binary spike events over a given number of time steps $T$, further detail can be found in the study[32].

### LFNL model
Federated learning has become an important paradigm aiming to train a collaborative AI model while keeping all the training data localized[13,14]. Thus, federated learning holds substantial promise for use in edge data analytics[15–17], which enables edge devices to train their AI models locally without sharing sensitive private data with external parties.

To perform model aggregation in LFNL, a set of edge devices $K$ participate in global neuromorphic model training (training federation for object recognition) with a leader to perform model aggregation. Each device $k$ adopts its local database $D_k$ to train its local neuromorphic model parameters $\boldsymbol{w}_k$ without sharing local data with the leader. In the federated learning scenario, a loss function $f(\boldsymbol{w}_k : \boldsymbol{x}_{k,j}, \boldsymbol{y}_{k,j})$ is introduced to quantify the federated performance error over the input data sample vector $\boldsymbol{x}_{k,j}$ on the training model $\boldsymbol{w}_k$ and the desired output scale vector $\boldsymbol{y}_{k,j}$ for each input sample $j$ at the $k$-th device. Accordingly, the local loss function on the training set $D_k$ at the $k$-th device can be expressed by

$$F(\boldsymbol{w}_k) = \frac{1}{|D_k|} \sum_{j \in D_k} f(\boldsymbol{w}_k : \boldsymbol{x}_{k,j}, \boldsymbol{y}_{k,j}), \tag{7}$$

where $|D_k|$ denotes the cardinality of the set $D_k$. At the leader, the global loss function with the local datasets of participating device

can be written as

$$F(\boldsymbol{w}) \triangleq \sum_{k \in K} \frac{|D_k|}{|D|} F(\boldsymbol{w}_k) = \frac{1}{|D_k|} \sum_{k \in K} \sum_{j \in D_k} f(\boldsymbol{w}_k : \boldsymbol{x}_{kj}, \boldsymbol{y}_{kj}), \qquad (8)$$

where $\boldsymbol{w}$ denotes the global model parameter at the leader and $D$ is the set of all data from all participating devices. The objective of the federated learning task is to find an optimal model parameter $\boldsymbol{w}^*$ by minimizing the global loss function[47] given by

$$\boldsymbol{w}^* = \arg\min F(\boldsymbol{w}). \qquad (9)$$

The leader iteratively updates the aggregated model through the local training procedure across edge devices in a group until the model converges to a certain learning accuracy target[48,49]. The leader election protocol and process can be found in Supplementary Fig. 1b, c.

### Energy consumption analysis

Similar to the work[50], the energy consumption can be estimated based on the number of floating point operations (FLOPS) of ANNs or SNNs which is approximately equivalent to the number of multiply-and-accumulate (MAC) operations. In the case of ANNs, FLOPS mainly consist of the MAC operations of convolutional and linear layers. On the contrary, for SNNs, as it performs training over binary spike signals, only accumulate (AC) operations are needed to handle the dot operations, except in the first input layer. For each convolutional layer with $I$ input of ANN or SNN, with $I$ input channels, $O$ output channels, $M \times M$ input feature map size, weight kernel size $p \times p$ and $Q \times Q$ output size, the number of FLOPS for ANNs and SNNs are respectively given by ref. 50

$$F^{ANN} = Q^2 \times I \times p^2 \times O, \qquad (10)$$

$$F^{SNN} = Q^2 \times I \times p^2 \times O \times R, \qquad (11)$$

where $R$ is the net spiking rate across training latency steps in SNNs, and we know $R < 1$ due to the sparse event-driven activity. We note that Eq. (11) is calculated over one time-step in SNN.

Based on the calculations from (10 and 11), we determine the total energy consumption by specifiying the energy consumption per MAC or AC operation on a 45 nm CMOS processor with 32-bit integer arithmetic[51]. Each MAC operation consumes 3.2 pJ while each AC operation needs only 0.1 pJ in the 45 nm CMOS processor. Hence, the total energy consumption for an ANN ($E_{ANN}$) and an SNN ($E_{SNN}$) can be calculated by ref. 50

$$E_{ANN} = \left( \sum_{l=1}^{L} F_l^{ANN} \right) \times E_{MAC}, \qquad (12)$$

$$E_{SNN} = F_1^{SNN} \times E_{MAC} + \left( \sum_{l=2}^{L} F_l^{SNN} \right) \times E_{AC} \times T, \qquad (13)$$

where $L$ is the number of layers of the ANN or SNN. $E_{MAC}$ and $E_{AC}$ are the energy consumption of one MAC or AC operation, respectively. In the case of SNNs, as shown in Eq. (13), calculating the total energy consumption requires considering the total number of AC operations over $T$ time training steps. In addition, the first layer (input layer) in SNNs needs to convert the analog input into binary spike events, and thus MAC operations are used in this layer. Note that the energy calculation in Eqs. (12) and (13) are approximate estimations which do not take into account the memory and any hardware circuit energy consumption.

### Leader election for LFNL

LFNL considers communication capability in leader election, as shown in Supplementary Fig. 1. In particular, it is preferred that the elected leader be located at the center of the edge devices, so that the communication capabilities (wireless communication link quality and Signal-to-Interference plus-Noise-Ratio (SINR)[52] from the followers to the leader are relatively balanced. In this context, taking the devices' communication capacity into account for the leader election can greatly improve the federated model aggregation speed in terms of lower latency.

### Experimental hardware and software

We implemented the experiments on two Raspberry PI 4Bs (CPU clock: 1.5 GHz, RAM: LPDDR4 8 GB), one Raspberry PI 3B+ (CPU clock: 1.4 GHz, RAM: LPDDR4 1 GB) and one laptop (CPU: 1.60 GHz, RAM: 8 GB). Note that when we implemented the experiment with more devices setting, we used the laptop. In terms of software, the experiments were performed using PyTorch via Python 3.0. The network and training parameters in our experiments are shown in Supplementary Table 1.

## Data availability

The sound or speech databases can be accessed at https://www.kaggle.com/vishnu0399/emergency-vehicle-siren-sounds. The image databases are collected from https://github.com/nikhilpatil99/Smart-Traffic-Management-Using-Deep-Learning, and https://www.kaggle.com/hj23hw/pedestrian-augmented-traffic-light-dataset[53], and the CIFAR10 and CIFAR100 datasets[44] are from (https://www.cs.toronto.edu/~kriz /cifar.html). The radar gesture database is collected from the study[30].

## Code availability

The code used in this research were developed using the Python and Matlab platform. To assist researchers in reproducing the experimental results, some code is available at https://github.com/GOGODD/FL-EDGE-COMPUTING/releases/tag/federated_learning. Some basic code was adopted from the studies[32,43].

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

## Acknowledgements

This research is supported by the National Research Foundation, Singapore under its Strategic Capability Research Centres Funding Initiative: Strategic Centre for Research in Privacy-Preserving Technologies & Systems; Nanyang Technological University (NTU) Startup Grant, Singapore Ministry of Education Academic Research Fund; the National Research Foundation, Singapore and Infocomm Media Development Authority under its Future Communications Research & Development Programme; the SUTD SRG-ISTD-2021-165; U.S. National Science Foundation under Grant CCF-1908308; Singapore Ministry of Education (MOE) Tier 1 (RG16/20); and National Natural Science Foundation of China under Grant U21A20444 and 61971366. Any opinions, findings and conclusions or recommendations expressed in this material are those of the author(s) and do not reflect the views of National Research Foundation, Singapore.

## Author contributions

H.L.Y., K.-Y.L., and L.X. conceived the ideal. H.L.Y. collected data, and designed the experiment. Z.H.X., H.H., N.D., and H.V.P. assisted to analyze the experiment results. All authors contributed to writing and revising the paper.

## Competing interests

The authors declare no competing interests.
