## [Peer Review File · Nature Communications]

REVIEWER COMMENTS

Reviewer #1 (Remarks to the Author):

This paper proposes a federated neuromorphic learning technique with resource and communication related constraints. Their leader election scheme allows them to elect one device with high capability (e.g., computation and communication capabilities) as a leader to manage the model aggregation, which effectively accelerates the federated learning speed and defends against model poisoning attacks. The authors show the effectiveness of their results on different audio, visual and radar datasets with uneven distribution.

The main issue I have with this work is novelty and technical contribution. Doing a google search on Federated spiking networks, I find the below paper [R1] that demonstrates federated learning in large-scale settings with up to 100-200 clients on more complex vision datasets like CIFAR100 (unlike this work) and show comprehensive analysis on different parameters like, straggler noise, # of clients, non-IID distribution among others.

I see that the authors have acknowledged this work as ref 43 in their paper, but when i see the places where the authors have referenced it, it seems bizarre and out of place. See below:

"We first tested the audio recognition capability of LFNL, and selected traffic sound dataset [43] for performance evaluation."

"The sound or speech databases can be accessed at <https://www.kaggle.com/vishnu0399/emergency-vehicle-siren-sounds>

[43]"

How is [43] related to any of the above two things? Even if i assume that the authors did an error in citation, they have not anywhere written about this most related work to theirs and have also not shown how their method compares to this work.

Further, if I look at the methods section, where the authors have done energy analysis, it seems like the entire calculation is similar to that of [R1] and it just accounts for analytical estimation of computations. Since the major and only difference i can find between the author's work and [43] is their leader election scheme that allows them to do aggregation based on client capability, I am wondering how will the energy calculation the authors talk about yields any benefits for communication or teh amount of memory required during aggregation etc..

Besides this major point, the paper is full of writing errors and honestly the datasets the authors have chosen like TIDIGITS, MNIST to showcase the efficacy is too small/trivial to even justify if their method is working.

I suggest the authors to read [R1], properly (both with quantitative and qualitative comparison) justify how their methodology differs from [R1], the type of approaches they use for instance ([R1] uses some batchnorm through time SNN training technique [R2]), justify how leader election leads to communication benefits with proper cost estimation methods accounting memory, and finally go for complex datasets with more clients!

[R1] Venkatesha, Yeshwanth, et al. "Federated learning with spiking neural networks." IEEE Transactions on Signal Processing 69 (2021): 6183-6194.

[R2] Kim, Youngeun, and Priyadarshini Panda. "Revisiting batch normalization for training low-latency deep spiking neural networks from scratch." Frontiers in neuroscience (2020): 1638.

Reviewer #2 (Remarks to the Author):

In this article, the authors present a technique, called lead federated neuromorphic learning (LFNL), that enables multiple edge devices to collaboratively train a global neuromorphic model without a coordinator. The technique effectively tackles the following two key challenges in the application of artificial intelligence (AI): 1) the limited local datasets of edge devices make accurate training of a learning model difficult; 2) AI model training is generally power hungry and impractical to be carried out on energy-constrained edge devices. Extensive experimental results are provided to validate the effectiveness of LFNL. The reviewer found the work interesting and novel and can find a wide range of practical AI powered applications that involves distilling distributed mobile data into intelligence.

With that said, the work should be improved in several aspects. The reviewer has the following comments.

1. In the proposed LFNL method, the global neuromorphic model is aggregated by collecting the local models from the participants. How should the global model aggregation be modified to address the issue of new participants joining or leaving the learning process? Is the LFNL is still effective under such circumstances? Some discussion addressing the issue would be useful.

2. The authors should highlight the benefits of having the leader in the LFNL framework. Currently, it is unclear how the leader is different from a central server.

3. The size of the training data is a key factor determining the classification accuracy. It is necessary to discuss whether the LFNL method can still achieve comparable testing accuracy, lower training latency, and less traffic compared with the conventional centralized learning if the training data size is not large such as in the case of MNIST and TIDIGITS datasets.

4. It would be more convincing if the authors could provide experimental results to test the classification accuracy robustness of LFNL on training data perturbed by noise.

5. On page 3, line 99, in Fig. 1f, there exists a situation where some participants have high-accuracy training models while other participants have low-accuracy training models. Please explain whether the global neuromorphic model would be more reliable if the leader only selects the high-accuracy participants to join the model aggregation?

6. On page 11, line 268, can the authors provide an explanation on why LFNL-SNN achieves a similar classification accuracy as LFL-ANN, but significantly reduces the energy consumption. Is this important for the application of neuromorphic engineering and whether LFNL-SNN can be flexibly applied in the domain of neuromorphic engineering?

Response Letter to the Reviewers

We are grateful for the valuable comments and suggestions for this manuscript (NCOMMS-22-01613) from all the reviewers. In the response below, comments from the reviewers are quoted in **black** font and is followed by the corresponding detailed response in **blue** font. We also have revised the manuscript and the supplementary information accordingly, and these updates are highlighted in **blue**.

Reviewer #1 (Remarks to the Author):

1. This paper proposes a federated neuromorphic learning technique with resource and communication related constraints. Their leader election scheme allows them to elect one device with high capability (e.g., computation and communication capabilities) as a leader to manage the model aggregation, which effectively accelerates the federated learning speed and defends against model poisoning attacks. The authors show the effectiveness of their results on different audio, visual and radar datasets with uneven distribution.

Response from Authors: We thank the reviewer for these positive comments. In the following, we carefully address all the comments point-by-point.

2. The main issue I have with this work is novelty and technical contribution. Doing a google search on Federated spiking networks, I find the below paper [R1] that demonstrates federated learning in large-scale settings with up to 100-200 clients on more complex vision datasets like CIFAR100 (unlike this work) and show comprehensive analysis on different parameters like, straggler noise, # of clients, non-IID distribution among others.

Response from Authors: We thank the reviewer for the valuable comments. The novelty and technical contribution of our work can be summarized as follows: This paper proposes a lead federated neuromorphic learning (LFNL), a decentralized brain-inspired computing method based on SNNs, enabling multiple edge devices to collaboratively train a global neuromorphic model without a central coordinator. In particular, we also present a leader election scheme to elect one device with high capability (e.g., computation and communication capabilities) as a leader to manage the model aggregation, which effectively accelerates the federated learning speed and defends against model poisoning attacks. The experimental results (Fig. 2 to Fig. 4 in the main manuscript) verify the effectiveness (lower data traffic size, lower training latency, i.e.) of the proposed LFNL method compared to other existing popular methods.

By contrast, the work in [R1] designed a centralized federated neuromorphic learning (CFNL) method for training decentralized and privacy preserving SNNs, where a central server is used to perform global SNNs model aggregation. CFNL can keep the raw data on the devices, and only the local model parameters need to be uploaded to the central server for model aggregation which enhances global accuracy. However, it relies on a centralized structure. In practical wireless mobile edge networks, due to the complex, dynamic and unreliable wireless communication environments, some devices may fail to communicate the gradients to the central server within a given target time limit. For example, due to the mobility of a part of wireless edge devices, the central server is not always at the center of the devices. In this case, as the central computing server is located at the edge or corner [R2, R3], some edge devices locating at the opposite corner have longer communication distances (i.e., a high path loss), resulting in a limited communication data rate and an increase in the data packet transmission latency. Thus, the central sever needs to wait for the local model parameters before performing model aggregation, which directly increases the overall processing delay and the probability of stragglers.

In order to verify the performance of LFNL in complex environments, we added the experimental demonstration of our lead federated neuromorphic learning (LFNL) technique in large-scale settings with more complex vision datasets (i.e., CIFAR10 and CIFAR100), which is evaluated under different parameters (i.e., participating device configurations, non-IID distribution, varied probability of stragglers and gradient noise). We conducted experiments by using VGG9 model on CIFAR10 and CIFAR100 datasets [R1], where a total of 50000 32×32 RGB images of training data and 10000 32×32 RGB images of testing data are used, respectively.

Fig. R1 illustrates the performance comparisons of our proposed LFNL method and the centralized federated neuromorphic learning (CFNL) method on CIFAR10 and CIFAR100 datasets with different participating device configurations. Note that LFNL has one leader to perform participating device selection and communication resource management, while the existing CFNL method does not adopt this technique. Here, similar to the study in [R1], we use the convention of P/N to show the device split, where N denotes the total number of devices (the dataset is divided into N parts for N devices), and P represents the number of participating devices selected for model aggregation in each global round. Other related parameters can be found in the reference [R1].

From Fig. R1, we can observe that the test accuracy of the two methods gradually decreases as the number of devices increases. The reason is that because the training dataset is divided among devices, the increasing number of devices will degrade the learning capacity with insufficient training dataset. Even though both of these methods show a similar trend under both training datasets, the proposed LFNL method achieves higher test accuracy than that of the CFNL method under different device convention factors. The reason is that LFNL enables the network to elect a leader with the high communication channel quality, and thus decreases the probability of stragglers during model aggregation, where the stragglers

represent that devices fail to communicate the model parameters (or gradients) with the central server or the leader within an acceptable time limit. By contrast, in CFNL, due to the mobility of edge devices, the quality of communication channels between some devices to the server is unreliable when the server is located at the edge of the devices with high channel path loss, leading to a higher probability of stragglers compared with LFNL, and eventually the server might receive fewer updates than expected.

Fig. R1: Performance evaluation of LFNL vs. CFNL for VGG9 model trained on CIFAR10 and CIFAR100 under different device configurations.

We further evaluated the classification accuracy of LFNL and CFNL under different degrees of the non-IID data distribution under CIFAR10 and CIFAR100, where each dataset is divided among 100 devices with 30 participating devices in each global round. Here, we use the Dirichlet distribution with the concentration parameter α to obtain non-identical datasets similar to the existing federated learning works [R1, R4, R5]. Note that as the value of α reduces, the class composition becomes more skewed and the degree of the data non-identical distribution becomes more severe. The corresponding results are provided in Fig. R2.

Fig. R2: Impacts of the non-IID factor on classification accuracy of LFNL vs. CFNL when the training dataset is divided among 100 devices and 30 devices participating in each global round.

From Fig. R2a, we found that even though the data becomes more non-IID as the parameter α decreases, the classification accuracies of both the LFNL and CFNL do not decline significantly when $\alpha \geq 1$. However, when $\alpha < 1$, the degree of non-IID in the training data tends to be more skewed as α decreases, and the training model diverges significantly, and lastly, there is a steady decrease in the classification accuracy performance of LFNL and CFNL. However, the classification accuracy of LFNL still outperforms CFNL regardless of different non-IID levels.

In practical mobile edge networks, due to the complex, dynamic and unreliable wireless communication environments, some devices may fail to communicate the gradients to the central server or the leader within a given target time limit. Thus, we evaluate the accuracy performance by changing the probability of one device failing to communicate the model parameters to the central server or the leader, which is defined as the probability of stragglers [R1]. As shown in Fig. R3, for both of CNFL and LFNL methods, the negative impacts of the probability of stragglers on the test accuracy are not significant in the case of IID training data. However, the accuracy decreasing trend becomes clear as the increase of the probability of stragglers in the case of non-IID training data. We would like to note that LFNL still achieves a higher classification accuracy than that of CNFL in different cases of stragglers (for probabilities < 0.5) under both CIFAR10 and CIFAR100 datasets, as the network elects one leader with high overall channel quality which reduces the probability of stragglers.

Fig. R3: Impacts of the probability of stragglers on the performance of LFNL vs CFNL when the training dataset is divided among 100 devices and 30 devices participating in each global round.

Considering that the gradients (or model parameters) may be obfuscated with added noise, the classification accuracies of LFNL and CFNL are evaluated with respect to the added noise in the gradients. Similar to the work in [R1], the Gaussian noise $\mathcal{N}(0,1)$ multiplied by noise strength is added in the model gradients of all participated devices. From Fig. R4, we can observe that the classification accuracies of the two methods degrade slightly with the increase of noise strength. The results indicate that both of the two

methods are robust to the noise in the model gradients. It is worth noting that LFNL still outperforms CFNL under different noise strengths in the gradients.

Fig. R4: Impacts of the gradients noise on the performance of LFNL vs CFNL when the training dataset is divided among 100 devices and 30 devices participating in each global round.

[R1] Venkatesha, Y., Kim, Y., Tassiulas, L. & Panda, P. Federated learning with spiking neural networks. *IEEE Trans. Signal Process.* **69**, 6183-6194 (2021).
[R2] Xu J., Wang, H. Client selection and bandwidth allocation in wireless federated learning networks: A long-term perspective. *IEEE Trans. Wireless Commun.* **20**, 1188-1200 (2021).
[R3] Yang, Z., Chen, M., Saad, W., Hong C. S., Shikh-Bahaei, M. Energy efficient federated learning over wireless communication networks. *IEEE Trans. Wireless Commun.* **20**, 1935-1949 (2021).
[R4] Wang, H. et al. Federated learning with matched averaging. 2020, arXiv:2002.06440.
[R5] Yurochkin, M. et al. Bayesian nonparametric federated learning of neural networks. in *Proc. Int. Conf. Mach. Learn.*, 2019.

3. I see that the authors have acknowledged this work as ref 43 in their paper, but when i see the places where the authors have referenced it, it seems bizarre and out of place. See below:

"We first tested the audio recognition capability of LFNL, and selected traffic sound dataset [43] for performance evaluation."

"The sound or speech databases can be accessed at <https://www.kaggle.com/vishnu0399/emergency-vehicle-siren-sounds>[43]"

How is [43] related to any of the above two things? Even if i assume that the authors did an error in citation, they have not anywhere written about this most related work to theirs and have also not shown how their method compares to this work.

Response from Authors: We thank the reviewer for this helpful suggestion. We made an error in the citation in the previously submitted version, where the citation needs to be revised as "The sound or speech databases can be accessed at <https://www.kaggle.com/vishnu0399/emergency-vehicle-siren-sounds> [44]" As suggested, we have modified the reference as [44] in the revised paper.

In the previously submitted version, we summarized the main contributions of this most related work [43] in ‘Section 1: Introduction’, i.e., “*Furthermore, several attempts have been made to combine SNNs with FL^{42,43} to improve both learning capability and energy efficiency, but model parameters are still aggregated by a central server*”. In addition, we have compared the experimental performances (i.e., test accuracy, data traffic size, and training time) of our proposed LFNL with this work [43] in previously submitted version, where the most related work [43] is similar to the compared centralized federated neuromorphic learning (CFNL) method, please see the results in Fig. 2, Fig. 3 and Fig. 4 in the main manuscripts. The principle of CFNL has been provided in Fig. 2c and Page 5 in the revised paper. In CFNL, the raw data is keep on the device sides (Fig. 2c), and only the local model parameters need to be uploaded to the central server for model aggregation which enhances the global accuracy, but CFNL relies on a centralized structure.

As shown in Fig. 2, Fig. 3 and Fig. 4 in the main manuscript, both the LFNL and CFNL methods have the comparable classification accuracies. However, as the CFNL method requires a central server which generates the higher data traffic size than that of LFNL in edge AI. It is worth noting that from Fig. 2, Fig. 3 and Fig. 4 in the main manuscripts, that the LFNL and CFNL methods have the similar training time when we assume that the central server in CFNL is at the center of the edge devices. However, in practical environments, the central server may not be always at the center of the edge devices, especially in the scenario that edge devices are moving. In this case, some edge devices may have a high channel path loss from them to the central server, which increases the probability of the stragglers.

As shown in Fig. R5, LFNL achieves the faster convergence speed than that of CFNL, because LFNL considers communication capacity into the leader election that results in the convergence speed enhancements, which supports real-time edge computing deployments.

Fig. R5: Convergence speed comparison of LFNL with leader election and CFNL with central server on traffic image dataset.

For example, as illustrated in Fig. R5, when the validation loss is 0.025, the training completion times of LFNL and CFNL are 20.6s and 34.3s, respectively, which indicates that

LFNL significantly reduces the overall training time by nearly 66.5%. The reason lies in the fact that electing the leader with high communication capabilities can substantially reduce communication latency, and improve the federated model aggregation speed. Hence, LFNL with leader election is valuable in accelerating the federated learning process on edge devices.

As suggested, we have added some necessary explanation about the most related work [43] to our proposed LFNL method, i.e., “The literature [43] designed a federated learning method for training decentralized and privacy preserving SNNs, where a central server is used to perform global SNNs model aggregation. We denote this method [43] as centralized federated neuromorphic learning (CFNL), which can keep the raw data on the device sides (Fig. 2c), and only the local model parameters need to be uploaded to the central server for the model aggregation and enhances the global accuracy, but it relies on a centralized structure.” The performance comparisons between LFNL and CFNL have been provided in Fig. 2-6 in the revised paper.

4. Further, if I look at the methods section, where the authors have done energy analysis, it seems like the entire calculation is similar to that of [R1] and it just accounts for analytical estimation of computations. Since the major and only difference I can find between the author's work and [43] is their leader election scheme that allows them to do aggregation based on client capability, I am wondering how will the energy calculation the authors talk about yields any benefits for communication or the amount of memory required during aggregation etc..

Response from Authors: Thank you very much for the valuable comments. The reference [R1] mentioned by the reviewer is Ref. [43] in our previously submitted paper, and we have updated it by using the following professional format in the revised paper:

[43] Venkatesha, Y., Kim, Y., Tassiulas, L. & Panda, P. Federated learning with spiking neural networks. *IEEE Trans. Signal Process.* **69**, 6183-6194 (2021).

The reviewer's comment is correct that the energy calculation in our work is similar to that of [R1], and some papers [R6-R10] also use this method to calculate energy consumption of ANN and SNN. The energy consumption estimation in our work has the following benefits or reasons in the federated model aggregation for edge artificial intelligence.

Limited energy and communication radio resource of edge devices are key challenges for deploying federated learning, as these factors directly affect the model aggregation performance. Energy consumption estimation can help edge devices to predict how much energy that they need to consume during model training or model aggregation. The energy prediction mechanism assists the mobile edge network to jointly optimize device leader election, device participating selection and communication bandwidth allocation given with long-term device energy constraints, which achieves long-term performance guarantee. For this benefit, the studies in [R2, R3, R11] have provided some simulation results to verify the effectiveness of resource scheduling by considering energy consumption in the federated

learning framework. Moreover, the leader is required to consume additional energy on global model aggregation and model parameters broadcasting. Energy consumption estimation can assist with the network to flexibly schedule the leader election, where a number of edge devices with rich energy can rotate become the leader to balance the energy consumption, while other devices with deficient energy can avoid consuming a considerable amount of energy as they are set to have less chance to be elected as a leader.

Hence, due to these resource constraints, it is necessary to consider the energy efficiency for FL implementation in wireless mobile networks.

- [R6] Pan, Z., Chua, Y., Wu, J., Zhang, M. & Li, H. & Ambikairajah E. An efficient and perceptually motivated auditory neural encoding and decoding algorithm for spiking neural networks. *Front. Neurosci.* **13**, 1420 (2020).
- [R7] Horowitz, M. Computing's energy problem (and what we can do about it). In Proc. *2014 IEEE International Solid-State Circuits Conference Digest of Technical Papers (ISSCC)*. 10–14 (IEEE, 2014).
- [R8] Priyadarshini, P., Aparna, A., Kaushik R., Toward scalable, efficient, and accurate Deep spiking neural networks with backward residual connections, stochastic softmax, and hybridization. *Front. Neurosci.* **14**, 1662 (2020).
- [R9] Han, S., Mao, H., and Dally, W. J. Learning both weights and connections for efficient neural network. *Advances in neural information processing systems*, 2015, pp. 1135 – 1143.
- [R10] Han, S., Mao, H., and Dally, W. J. Deep compression: Compressing deep neural networks with pruning, trained quantization and huffman coding. arXiv preprint arXiv:1510.00149, 2015.
- [R11] Chen, M., Yang, Z., Saad, W., Yin, C., Poor H. V., Cui, S. A joint learning and communications framework for federated learning over wireless networks. *IEEE Trans. Wireless Commun.* **20**, 269-283 (2021).

5. Besides this major point, the paper is full of writing errors and honestly the datasets the authors have chosen like TIDIGITS, MNIST to showcase the efficacy is too small/trivial to even justify if their method is working.

Response from Authors: We thank the reviewer for this insightful comment. As suggested, in the revised paper, we have carefully checked the English usage and tried our best to correct the grammatical mistakes and writing errors for the better quality of the paper.

As suggested, according to [R1], we have selected more complex datasets including CIFAR10 and CIFAR100 to examine the performance of our proposed method. Moreover, we have rigorously evaluated the impacts from various parameters including the probability of straggler, the gradient noise, the number of participating devices and the non-IID distribution, which have been provided in the response to *Comment 1*. According to the experimental results (Fig. R1 to Fig. R4), we can observe that the proposed LFNL can work well on the CIFAR10 and CIFAR100 datasets. In addition, we have added the results and analysis in Fig. 6 in the revised main manuscript with red color.

6. I suggest the authors to read [R1], properly (both with quantitative and qualitative comparison) justify how their methodology differs from [R1], the type of approaches they use for instance ([R1] uses some batchnorm through time SNN training technique [R2]), justify how leader election leads to communication benefits with proper cost estimation methods accounting memory, and finally go for complex datasets with more clients!

[R1] Venkatesha, Yeshwanth, et al. "Federated learning with spiking neural networks." IEEE Transactions on Signal Processing 69 (2021): 6183-6194.

[R2] Kim, Youngeun, and Priyadarshini Panda. "Revisiting batch normalization for training low-latency deep spiking neural networks from scratch." Frontiers in neuroscience (2020): 1638.

Response from Authors: Thank you very much for the reviewer's insightful comments. As suggested, we have carefully read these two studies [R1, R2] and summarized the contributions of these two references in the revised paper. Leader election has the following communication benefits with proper cost estimation methods.

Fig. R6: a) An example of the distributions of mobile edge devices and the central server in two time slots under the CNL-based framework. b) An example of the distributions of mobile edge devices and the leader in two time slots under the LFNL-based framework.

Leader election can decrease the overall communication latency and reduce the probability of stragglers during model aggregation, where the straggler represents that devices fail to communicate the model parameters (or gradients) to the central server or the leader. Fig. R6a shows the distributions of the central computing server and wireless edge devices in

the centralized federated neuromorphic learning (CFNL) framework, while Fig. R6b depicts the distributions of the elected leader and wireless edge devices in the lead federated neuromorphic learning (LFNL) framework. Due to the mobility of a part of wireless edge devices, the central server is not always at the center of the devices. In this case, as the central computing server is located at the edge or corner, some edge devices locate at the opposite corner have longer communication distances (i.e., a high path loss), resulting in a limited communication data rate and an increase in the data packet transmission latency. Thus, the central computing sever needs to wait for these devices to upload the local model parameters before performing model aggregation, which directly increases the overall processing delay. Moreover, due to the severe path loss from these devices to the central computing sever, it is challenging for these devices to successfully send their updates to the server within an acceptable time limit. Thus, the server might end up with fewer updates than expected [R1].

On the contrary, this challenge can be effectively addressed by considering the communication capability in the process of leader election, as shown in Fig. R6b. The reason is that the elected leader locates near the center of the edge devices, and the communication distance, which directly affects the wireless communication link quality, from edge devices to the leader are relatively balanced, and thus there does not exist an extreme situation in which the communication distance between any edge follower and the leader is extremely far. In this context, taking the device's metrics into account for the leader election can greatly improve the federated model aggregation performance in terms of lower latency and the lower probability of stragglers.

It is worth noting that the leader election needs to know the channel state information or the locations of wireless edge devices, which is the cost of our proposed method. This cost is affordable and justifiable, where devices share its locations in the network by wireless communications. This communication cost also exists in the general federated learning framework [R1], in which devices need to estimate the channel state information from them to the central computing server before sending the model parameters or gradients.

Here, we provide the performance evaluation of the proposed LFNL method on complex datasets (CIFAR10 and CIFAR100 datasets) with more devices, which are shown in Fig. R7. Note that the classification accuracy under different parameters (i.e., the device configuration, the probability of straggler, the gradient noise and the non-IID distribution) on CIFAR10 and CIFAR100 datasets with large devices have been provided in Fig. R2 to Fig. R4.

We evaluated the impacts of the number of devices participating on the classification accuracy of LFNL and CFNL on CIFAR10 and CIFAR100 datasets, as the number of participants also plays a key role in the accuracy performance in global model aggregation. As shown in Fig. R7a, in case of IID CIFAR10 dataset, both LFNL and CFNL are preserving the classification accuracy as the number participating devices decrease. However, there exists a gradual accuracy decline as the number of participating devices reduces in case of non-IID datasets in both LFNL and CFNL. Interestingly, as illustrated in Fig. R7b, for the LFNL and CFNL methods, the classification accuracy of both IID and non-IID cases have the

similar decrease trend as the number of participating devices decreases. The results in Fig. R8a also indicate that the classification accuracy of LFNL is higher than that of CFNL under different situation, and the advantage is more obvious when the number of participating devices is relatively small.

Fig. R7: Impacts of the number of participated devices on the performance of LFNL vs. CFNL on CIFAR10 and CIFAR100 when the training dataset is divided among 100 devices.

Reviewer #2 (Remarks to the Author):

In this article, the authors present a technique, called lead federated neuromorphic learning (LFNL), that enables multiple edge devices to collaboratively train a global neuromorphic model without a coordinator. The technique effectively tackles the following two key challenges in the application of artificial intelligence (AI): 1) the limited local datasets of edge devices make accurate training of a learning model difficult; 2) AI model training is generally power hungry and impractical to be carried out on energy-constrained edge devices. Extensive experimental results are provided to validate the effectiveness of LFNL. The reviewer found the work interesting and novel and can find a wide range of practical AI powered applications that involves distilling distributed mobile data into intelligence.

With that said, the work should be improved in several aspects. The reviewer has the following comments.

Response from Authors: We thank the reviewer for these positive comments. In the following, we fully address all the comments point-by-point.

1. In the proposed LFNL method, the global neuromorphic model is aggregated by collecting the local models from the participants. How should the global model aggregation be modified to address the issue of new participants joining or leaving the learning process? Is the LFNL

is still effective under such circumstances? Some discussion addressing the issue would be useful.

Response from Authors: Thank you very much for the reviewer's valuable comments. If a new participant joins in the LFNL-based system, transfer learning can be used to integrate the global model with the model of the new participant, as shown in Fig. R9. Transfer learning is a powerful AI tool for improving learning efficiency by transferring the learned knowledge or trained model from one related task to another new task.

As shown in Fig. R8a, the leader in the LFNL-based system can transfer the existing global model to the new participant, where a new model is generated for the new participant by integrating the global model and the new participant's previous model. After that, the new participant adopts the integrated model for its local model training before joining the next global model aggregation. It is worth mentioning that if the new participant does not have the training model, as shown in Fig. R8b, it can directly use the global model as its personal model for the local model training. In our future work, we will consider transfer learning into our proposed LFNL framework to improve the scalability and flexibility.

Fig. R8: Transfer learning for the LFNL-based system. a) An example of the model integration of the existing global model and the personal local model of the new participating device. b), An example of transferring the existing global model to the new participating device.

2. The authors should highlight the benefits of having the leader in the LFNL framework. Currently, it is unclear how the leader is different from a central server.

Response from Authors: Thank you very much for the reviewer's insightful suggestion. Compared with the centralized federated neuromorphic learning in the presence of a fixed central server (without leader election) at a base station or an access point, leader selection in LFNL is dynamic and is able to adapt to the network and energy states of the devices in the network. The benefits of having the leader in the LFNL framework are listed as follows:

1) Decreased traffic data size: For the centralized solutions in the centralized federated neuromorphic learning (CFNL), the model parameters are still kept by the central server. Furthermore, such star-shaped architectures decrease fault tolerance and increase the data traffic size. By contrast, having the leader in LFNL can significantly decrease the traffic data size, which have been shown in Fig. 2m in the main manuscript and Supplementary Fig. 7e in supplementary information document.

2) Enhanced federated aggregation process and reduced overall training latency: Leader election plays an important role in federated model aggregation performance for edge AI. The leader with high computation and communication capabilities can speed up the federated aggregation process and reduce the overall training latency, whereas the federated aggregation latency will be increased if the network elects a leader with low computation and communication capabilities. For example, as illustrated in Supplementary Fig. R9, there are three scenarios in terms of communication capability. We consider the communication capability as an example, and assume that other constraints (e.g., computation and energy supply capabilities) are the same for all devices. Note that with the increase of path loss, the communication data rate decreases [R12-R14], thereby leading to the increase of data packet transmission latency. For Scenario 1 or Scenario 2, the central server is very far from some edge devices, the followers (e.g., device 2) located at the opposite corner have a greater communication distance, resulting in a limited communication data rate and increasing the data packet transmission latency.

Fig. R9: An example of the central server and leader election in terms of communication capability.

On the contrary, this problem can be effectively addressed by considering the communication capability into the leader election, as shown in Scenario 3 (Fig. R9). The reason is that the elected leader locates at the center of the edge devices, and the communication distance (wireless communication link quality) from followers to the leader are relatively balanced, and thus there does not exist the extreme situation where the

communication distance between any edge follower and the leader is extremely far. In this context, taking the device's metrics into account for the leader election can greatly improve the federated model aggregation in terms of lower latency.

The document of *Supplementary Information* provides the performance comparisons between the proposed leader election method and the CFNL method with central server. Fig. R10 shows the convergence speed of the two methods. Overall, the proposed LFNL method considers communications capacity into the leader election that results in the convergence speed enhancements. In particular, we find that LFNL obtains the faster convergence speed while maintaining the lower training loss via training time slots. For example, as illustrated in Fig. R10, when the validation loss is 0.025, the training completion times of the LFNL and the CFNL methods are 20.6s and 34.3s, respectively. The LFNL method significantly reduces the overall training time by nearly 66.5%. The reason lies in the fact that electing the leader with high communication capabilities to perform federated model aggregation, which substantially reduces computation and communication times. Hence, LFNL with leader election is valuable in accelerating the federated learning process on edge devices.

Fig. R10: Validation loss vs. training time for the LFNL method with leader election and the CFNL method with central server on traffic image dataset [R15, R16].

[R12] Sjöberg A., Gustavsson E., Koppisetty A.C., & Jirstrand M. Federated learning of deep neural decision forests. machine learning, optimization, and data science. *Lecture Notes in Computer Science*, 11943 (2019).

[R13] Yang, Z., Chen, M., Saad, W., Hong C. & Shikh-Bahaei, M. Energy efficient federated learning over wireless communication networks. *IEEE Trans. Wireless Commun.* **20**, 1935 – 1949 (2021).

[R14] Yang, H., Zhao, J., Xiong, Z., Lam, K. -Y., Sun, S., & Xiao, L. Privacy-preserving federated learning for UAV-enabled networks: Learning-based joint scheduling and resource management. *IEEE J. Sel. Areas Commun.* **39**, 3144 – 3159 (2021).

[R15] Chintamani, N., Yash, M., Nikhil, P., Vivek, P. & Sandeep, P. Smart traffic control using deep learning. Preprint at <https://github.com/nikhilpatil99/Smart-Traffic-Management-Using-Deep-Learning> (2019).

[R16] <https://www.kaggle.com/hj23hw/pedestrian-augmented-traffic-light-dataset>.

3. The size of the training data is a key factor determining the classification accuracy. It is necessary to discuss whether the LFNL method can still achieve comparable testing accuracy, lower training latency, and less traffic compared with the conventional centralized learning if the training data size is not large such as in the case of MNIST and TIDIGITS datasets.

Response from Authors: Thank you for the reviewer's insightful suggestion. As suggested, we provide the performance comparisons between the LFNL method and conventional centralized learning (called centralized neuromorphic learning: CNL), i.e., classification accuracy, training latency, and traffic size. Fig. R11 shows the performance evaluations of the two methods on MNIST and TIDIGITS datasets, where we use the 20% training data size of MNIST dataset and the 30% training data size of TIDIGITS dataset. The training dataset are divided into three parts for three edge devices under a Non-IID data distribution. From Fig. R11, we can observe that the classification accuracy of LFNL slightly has a slightly larger loss (up to 1 % loss) compared to CNL, but it achieves significantly the lower traffic size and training latency than those of CNL. For example, when we train the model on MINIST dataset, the data traffic load and training latency can be respectively reduced by 69.63% and 81.18% by using the LFNL method, respectively. According to these results, we can conclude that the LFNL method can still achieve the comparable testing accuracy, lower training latency, and less traffic data size compared with the conventional centralized learning in the case of MNIST and TIDIGITS datasets.

Fig. R11: Classification evaluation on MNIST and TIDIGITS datasets. a-c, Classification accuracy, traffic size, and training latency performance comparisons of LFNL and CNL on MNIST dataset, where we use 20% of MNIST dataset for training. d-f, Classification accuracy, traffic size, and training latency performance comparisons of LFNL and CNL on TIDIGITS dataset, where we use 30% of TIDIGITS dataset for training.

We have added these results and analysis in ‘‘Supplementary Information, Subsection SI.5’’.

4. It would be more convincing if the authors could provide experimental results to test the classification accuracy robustness of LFNL on training data perturbed by noise.

Response from Authors: Thank you for the reviewer’s valuable comments. We evaluate the robustness of LFNL by adding noise into the model gradients and the raw datasets.

Case I: We evaluate the change in classification accuracy by adding different levels of noise into the model gradients. Similar to the work in [R1], the Gaussian noise $\mathcal{N}(0,1)$ multiplied by noise strength is added in the model gradients of all participated devices. We performed experiments by using the VGG9 model on CIFAR10 and CIFAR100 datasets, where a total of 50000 32×32 RGB images of training data and the 10000 32×32 RGB images of testing data are used. From Fig. R12, we can observe that the classification accuracy of the two methods degrades slightly with the increase of noise strength. The results indicate that both of the two methods are robust to the adding noise in the model gradients.

Fig. R12: Impacts of the gradient noise on the performance of LFNL vs CFNL when the training dataset is divided among 100 clients and 30 clients participating in each global round.

Case II: We then evaluate the change of classification accuracy by adding Gaussian noise $\mathcal{N}(0, \delta^2)$ to the raw datasets. Here, we select the traffic sound dataset [R17] as an example to evaluate the classification accuracy. Let P denote the signal power of input sound signal,

let δ^2 denote the added Gaussian noise power, and then the signal-to-noise ratio (SNR) is defined as $SNR = 10 \log_{10}(P / \delta^2)$ in dB. Note that the lower SNR is, the greater strength of the added Gaussian noise is. From Fig. R13, we can observe that the classification accuracies of the two methods slightly decline as the SNR decreases when $SNR \geq 15\text{dB}$, because the added Gaussian noise is not large in this region. However, when the SNR is lower than a certain level, i.e., $SNR < 15\text{dB}$, the two methods suffer from a large classification accuracy decrease at the higher noise intensity. Under different noise intensities, the two methods have comparable classification accuracy performance and similar robustness.

Fig. R13: Performance change with respect to the strength of the Gaussian noise added in the training data.

[R17] <https://www.kaggle.com/vishnu0399/emergency-vehicle-siren-sounds>.

5. On page 3, line 99, in Fig. 1f, there exists a situation where some participants have high-accuracy training models while other participants have low-accuracy training models. Please explain whether the global neuromorphic model would be more reliable if the leader only selects the high-accuracy participants to join the model aggregation?

Response from Authors: Thanks for the reviewer's comment. There are mainly two cases where the leader election the high-accuracy participants only to join the model aggregation may have opposite outcomes.

Case I: Due to the Non-IID dataset distribution among edge devices, in the first few several global learning epochs, the participants with sufficient datasets have high classification accuracy while other participants with insufficient datasets have low classification accuracy. If the leader only selects the high-accuracy participants to join the model aggregation, the training model parameters of the low-accuracy participants cannot be aggregated in the global model, and finally the features of the low-accuracy participants' datasets fail to be contributed

to the global model. The global aggregated training model cannot learn the overall features of all participants' datasets, and thus its classification accuracy is lower than that of the global model if the leader selects all participants to join the model aggregation. Based on the above analysis, the global neuromorphic model would be more reliable if the leader selects all participants to join the model aggregation.

Case II: When the training datasets or models of some participants contain noise, the low-quality model parameters will degrade the global model classification accuracy if the leader aggregates these training models into the global model. In this case, the global neuromorphic model will be more reliable if the leader only selects the high-accuracy participants to join the model aggregation, and prevents low-quality devices from affecting the learning accuracy.

According to the aforementioned discussion, whether the leader should select only the high-accuracy participants to join the model aggregation depends on the predefined system and environment configurations.

6. On page 11, line 268, can the authors provide an explanation on why LFNL-SNN achieves a similar classification accuracy as LFL-ANN, but significantly reduces the energy consumption. Is this important for the application of neuromorphic engineering and whether LFNL-SNN can be flexibly applied in the domain of neuromorphic engineering?

Response from Authors: Thanks for the reviewer's helpful suggestion.

On page 11, note that both SNNs and ANNs have the same learning structure for fair comparison. Fig. R14 provides the typical neuron models of ANNs and SNNs, where SNNs have a similar structure but different behavior compared with the ANN neuron [R18]. In ANN, neuron with each other adopts coded activation in high-precision as well as continuous values, and propagates signals in the spatial region. Different from ANNs, SNNs show signal in spike sequences by binary events instead of continue activation and each spiking neuron goes through rich dynamic behaviors. Thus, the presented LFNL-SNN generally has more temporal versatility and achieves the comparable classification accuracy (as shown Figs. 5a, 5d, and 5g in the main manuscript) compared to LFNL-ANN which mainly has spatial propagation and continuous activation [R19, R20].

Since a spike only fires when the membrane potential exceeds a threshold, the entire spike signals are often sparse and the signal can be event driven (only enabled when a spike input arrives). Furthermore, because the spike is binary (i.e., 0 or 1), the costly multiplication between the input and weight can be removed if the integration time window T is equal to 1 [R18]. For the above reasons, LFNL-SNN can usually achieve lower power consumption compared to LFL-ANN with intensive computation [R18-R20].

Fig. R14: Basic learning models of (a) ANNs and (b) SNNs [R18]

The above characteristics (comparable classification accuracy and lower energy consumption compared to common deep learning models) are important for the applications of neuromorphic engineering, and LFNL-SNN can be flexibly applied in the domain of neuromorphic engineering. Owing to the benefit of low energy consumption, LFNL can effectively deploy deep learning of neural networks for resource-constrained edge devices with various practical applications, such as speech recognition, image and video classification, smart sensing, health monitoring and multiple object detection in edge AI. In brief, our demonstration of the LFNL-SNN model is a step forward towards achieving energy efficient and bio-realistic neuromorphic hardware. Due to the decentralized federated learning and parallel training structures of LFNL, the LFNL-SNN directly replaces the centralized data sharing paradigm across edge device without any central server, and thus significantly reduces the heavy data traffic, and also enforces data privacy and decreases training latency compared to existing centralized learning methods. Moreover, with the implementation of spike-based processing features in LFNL, our platform is highly economic in energy, which makes LFNL available to energy-constrained edge devices. This gives us the confidence that a more specialized neuromorphic implementation of our model represents an alternative to current solutions based on LFNL-SNN architectures, especially in edge computing scenarios.

[R18] Deng, L., Wu, Y. J., Hu, X., Liang, L., Ding, Y. F., Li, G. Q., Zhao, G. S., Li, P., Xie, Y. Rethinking the performance comparison between SNNs and ANNs. *Neural Networks* **121**, 294-307 (2020).

[R19] Gerstner, W., Kistler, W. M., Naud, R., & Paninski, L. *Neuronal dynamics: From single neurons to networks and models of cognition*. Cambridge University Press. (2014).

[R20] Subbulakshmi Radhakrishnan, S., Sebastian, A., Oberoi, A., Sarbashis, D., & Saptarshi D., A biomimetic neural encoder for spiking neural network. *Nat. Commun.* **12**, 2143 (2021).

REVIEWER COMMENTS

Reviewer #1 (Remarks to the Author):

Thanks for the detailed response. The revised paper has addressed all my concerns. I think the paper has strengthened a lot with more qualitative and quantitative comparison.

I have some minor suggestions for the authors:

1) I spotted a few grammatical errors while reading the paper in a couple of places. Please correct them.

2) In Fig. 6 in main manuscript, where authors have compared between LFNL and CFNL [43], if the authors have not re-run the CFNL experiments themselves, I would suggest the authors to write in the figure caption that the "CFNL results were taken from Venkatesha et al. [43]" so that it is clear for the readers later who want to reproduce the results. If the authors have re-run the experiments, then the authors should mention that "the authors used the github repository provided by Venkatesha et al. [43] to get the CFNL results." Also, in the legends, please mention CFNL[43] so that readers can easily capture that CFNL is from another work.

Overall, I think the paper has improved substantially and deems publication.

Reviewer #2 (Remarks to the Author):

I appreciate the authors' effort on providing detailed and comprehensive explanations and conducting additional experiments to address my comments. Most of the concerns arising from my reading of this work have been cleared. My only additional comment is on the claimed limitations of the conventional federated learning scheme using an edge server as a way to argue the advantages of the proposed design, which uses an elected device in place of the server. Specifically, the former is claimed to suffer from long propagation distance, incurring low communication rates and long learning latency. The claims may not be fair as an edge server is typically located in proximity of devices. Furthermore, edge servers, which are connected to backhaul networks, can enable large-scale hierarchical federated learning to leverage more distributed data as well as facilitating the sharing of the trained model by network users. Such functions are out of reach by the proposed

scheme targeting only a cluster of devices. Thus it is suggested that the comparison with conventional schemes to be revised to be fairer. Other than that, I deem this work making a novel contribution and can find applications in many intelligent Internet-of-Things applications and thus recommend its publication.

Response Letter to the Reviewers

We are grateful for the valuable comments and suggestions for this manuscript (NCOMMS-22-01613A) from all the reviewers. In the response below, comments from the reviewers are quoted in **black** font and are followed by the detailed responses in **blue** font. We also have revised the manuscript and the supplementary information accordingly, and these updates are highlighted in **blue**.

Reviewer #1 (Remarks to the Author):

Thanks for the detailed response. The revised paper has addressed all my concerns. I think the paper has strengthened a lot with more qualitative and quantitative comparison.

Response from Authors: We thank the reviewer for these positive comments. In the following, we address the two comments point-by-point.

I have some minor suggestions for the authors:

1) I spotted a few grammatical errors while reading the paper in a couple of places. Please correct them.

Response from Authors: We thank the reviewer for the suggestion. We carefully reviewed the paper and corrected the grammatical mistakes and writing errors to improve the quality of the paper.

2) In Fig. 6 in main manuscript, where authors have compared between LFNL and CFNL [43], if the authors have not re-run the CFNL experiments themselves, I would suggest the authors to write in the figure caption that the "CFNL results were taken from Venkatesha et al. [43]" so that it is clear for the readers later who want to reproduce the results. If the authors have re-run the experiments, then the authors should mention that "the authors used the github repository provided by Venkatesha et al. [43] to get the CFNL results." Also, in the legends, please mention CFNL[43] so that readers can easily capture that CFNL is from another work.

Response from Authors: Thank you very much for the valuable comments. In Fig. 6 in the main manuscript, we re-run the experiments of the CFNL results by using the python codes from github provided by Venkatesha et al. [43]. Thus, as suggested, we have added the sentence "We used the github repository provided by Venkatesha et al. [43] to run the experiments of the CFNL results" in the revised version. In addition, we also added "CFNL[43]" in the legends of Fig. 6 to remind readers that CFNL is from another work.

Fig. 6: Classification accuracy evaluation on CIFAR10 and CIFAR100 datasets. a, b, Performance evaluation of LFNL vs CFNL under different device configurations. **c, d,** Impact of the non-IID data factor on the classification accuracy of LFNL vs CFNL when the training dataset is divided among 100 devices and 30 devices participating in each global round. **e, f,** Effects of the probability of stragglers on the performance of LFNL vs CFNL when the training dataset is divided among 100 devices and 30 devices participating in each global round.

3) Overall, I think the paper has improved substantially and deems publication.

Response from Authors: We thank the reviewer for recommending the publication of this work.

Reviewer #2 (Remarks to the Author):

I appreciate the authors' effort on providing detailed and comprehensive explanations and conducting additional experiments to address my comments. Most of the concerns arising from my reading of this work have been cleared. ① My only additional comment is on the claimed limitations of the conventional federated learning scheme using an edge server as a way to argue the advantages of the proposed design, which uses an elected device in place of the server. Specifically, the former is claimed to suffer from long propagation distance, incurring low communication rates and long learning latency. The claims may not be fair as an edge server is typically located in proximity of devices. ② Furthermore, edge servers, which are connected to backhaul networks, can enable large-scale hierarchical federated learning to leverage more distributed data as well as facilitating the sharing of the trained model by network users. Such functions are out of reach by the proposed scheme targeting only a cluster of devices. Thus it is suggested that the comparison with conventional schemes to be revised to be fairer. Other than that, I deem this work making a novel contribution and can find applications in many intelligent Internet-of-Things applications and thus recommend its publication.

Response from Authors: We thank the reviewer for these positive comments and valuable suggestions.

For comment ①:

We agree with the reviewer's point that an edge server is typically located in the proximity of devices when edge devices are not dynamically moving. However, as edge devices may move in practical environments, the edge server is unnecessary to be located in proximity of devices. In our work, we assume that the edge server is fixed, such as the base station or access point, which cannot move. Fig. R1a shows the distributions of the edge server and wireless edge devices in the centralized federated neuromorphic learning (CFNL) framework, while Fig. R2b depicts the distributions of the elected leader and wireless edge devices in the lead federated neuromorphic learning (LFNL) framework. Due to the mobility of a part of wireless edge devices, the edge server is not always at the center of the devices. In this case, as the edge server is located at the edge or corner, some edge devices locate at the opposite corner have longer communication distances (i.e., a high path loss), resulting in a limited communication data rate and an increase in the data packet transmission latency. Moreover, due to the severe path loss from these devices to the edge sever, it is challenging for these devices to successfully send their updates to the server within an acceptable time limit. Thus, the server might end up with fewer updates than that expected [R1].

Fig. R1: a) An example of the distributions of mobile edge devices and the edge server in two time slots under the CFNL-based framework. b) An example of the distributions of mobile edge devices and the leader in two time slots under the LFNL-based framework.

On the contrary, this challenge can be effectively addressed by considering the communication capability in the process of leader election, as shown in Fig. R2b. The reason is that the elected leader locates near the center of the edge devices, and the communication distance, which directly affects the wireless communication link quality, from edge devices to the leader are relatively balanced. In this case, there does not exist an extreme situation in which the communication distance between any edge follower and the leader is extremely far. In this context, taking the device's metrics into account for the leader election can greatly improve the federated model aggregation performance in terms of lower latency and the lower probability of stragglers, where the stragglers represent that devices fail to communicate the model parameters (or gradients) with the central server or the leader within an acceptable time limit.

In Fig. 2, Fig. 3, and Fig. 4 in the main manuscript, for fair comparisons of performance, we have set that the edge server is located at the central place of the wireless edge devices, and thus we can observe that the centralized federated neuromorphic learning (CFNL) and our proposed scheme (LFNL) have the similar training latency (shown in Fig. 2n, Fig. 3j, and Fig. 4h in the manuscript).

In order to verify additional advantages of the proposed scheme, in Fig. 6 in the manuscript and Supplementary Fig. 2, we assume that the edge server is not always present at the central place of the wireless edge devices and sometimes it locates at the corner (at the left

figure of Fig. R1a) of wireless edge devices due to the dynamic movements of devices. In this case, as shown in Supplementary Fig. 2, the proposed scheme (LFNL) achieves the faster convergence speed due to the low communication latency. In addition, as shown in Fig. 6 in the manuscript, the LFNL method achieves higher test accuracy than that of the CFNL method. The reason is that LFNL enables the network to elect a leader with the high communication channel quality, and thus decreases the probability of stragglers during model aggregation.

For comment ②:

The reviewer's claim is professional that when edge servers are connected to backhaul networks, these edge servers can enable large-scale hierarchical federated learning to leverage more distributed data as well as facilitating the sharing of the trained model by network devices. As our work investigates the scenario that a number of wireless edge devices perform federated learning in a cluster framework, and thus this function was not studied in the proposed scheme in the previously submitted version. For fair comparison, if we set that a number edge servers are connected to backhaul networks to perform large-scale hierarchical federated learning, we also need to set that there have a number of leaders in the edge networks. In this context, in our proposed LFNR framework, edge devices are divided into a number of groups with each group being with one elected leader. Since this is irrelevant to the selling point in this work, we will explore this part in the future work.

[R1] Venkatesha, Y., Kim, Y., Tassiulas, L. & Panda, P. Federated learning with spiking neural networks. *IEEE Trans. Signal Process.* **69**, 6183-6194 (2021).

REVIEWERS' COMMENTS

Reviewer #1 (Remarks to the Author):

The authors have addressed my comments.

Reviewer #2 (Remarks to the Author):

The authors' clarification and explanation are sufficient to address my concerns. I have no more comments.

Response Letter to the Reviewers

We are grateful for the valuable comments and suggestions for this manuscript (NCOMMS-22-01613B) from all the reviewers. In the response below, comments from the reviewers are quoted in **black** font and are followed by the detailed responses in **blue** font.

Reviewer #1 (Remarks to the Author):

The authors have addressed my comments.

Response from Authors: We thank the reviewer for these positive comments.

Reviewer #2 (Remarks to the Author):

The authors' clarification and explanation are sufficient to address my concerns. I have no more comments.

Response from Authors: We thank the reviewer for these positive comments.